# `PropMEND`: Hypernetworks for Knowledge Propagation in LLMs

## Abstract

Knowledge editing techniques for large language models (LLMs) can inject knowledge that is later reproducible verbatim, but they fall short on *propagating* that knowledge: models cannot answer questions that require them to reason with the injected knowledge. We present a hypernetwork-based approach for knowledge propagation, where we meta-learn how to modify gradients of a language modeling loss to encourage injected information to propagate. Our approach, `PropMEND`, extends the meta-objective of MEND [29] so that gradient updates on a piece of knowledge are transformed to allow answering of multi-hop questions involving that knowledge. On the `RippleEdit` dataset, our method significantly improves performance on propagation questions whose answers are not explicitly stated in the injected fact, in contrast to existing methods that only improve on propagation questions where the answer can be copied verbatim. To study the extent of generalization that our propagation achieves, we construct `StoryPropagation`, a controlled dataset focusing on entities and relations that the model already understands well. We find that `PropMEND` generalizes effectively to partially unseen entity-relation pairs, indicating the effectiveness of our meta-trained hypernetwork for knowledge propagation.

## 1  Introduction

Knowledge editing methods [26; 29; 7; 37] show strong performance in transforming large language models (LLMs) to *reproduce* injected knowledge, but induce very limited *propagation* of that knowledge [6; 46]. This failure stands in disappointing contrast to LLMs' ability to propagate knowledge that is given in context at inference time [31; 45]. Although propagation can be improved through training on substantially more data [33; 1; 3], these methods do not provide an efficient way to inject knowledge, requiring large-scale data augmentation for each knowledge to be injected [42].

In this work, we propose a new knowledge editing approach, named `PropMEND`, that achieves substantially improved results at knowledge propagation. Our method builds upon Model Editor Networks using Gradient Decomposition (MEND) [29], which introduces auxiliary hypernetworks to make efficient, local edits to LMs. We propose to train these hypernetworks with knowledge propagation as the core objective. Taking in a model's gradient from the language modeling objective on the injected fact as input, we train hypernetworks to modify that gradient to enable LMs to answer propagation questions involving that fact correctly when the output gradient is applied; see Figure 1. We further identify that hyperparameters (e.g., layers in which model updates are applied) impact the propagation performance significantly.

We first evaluate our approach on `RippleEdit` [6], a knowledge propagation question answering dataset. We identify existing methods that only excel in instances where the target answer appears verbatim in the injected facts, while achieving negligible improvement on non-verbatim questions. We

Submitted to 39th Conference on Neural Information Processing Systems (NeurIPS 2025). Do not distribute.

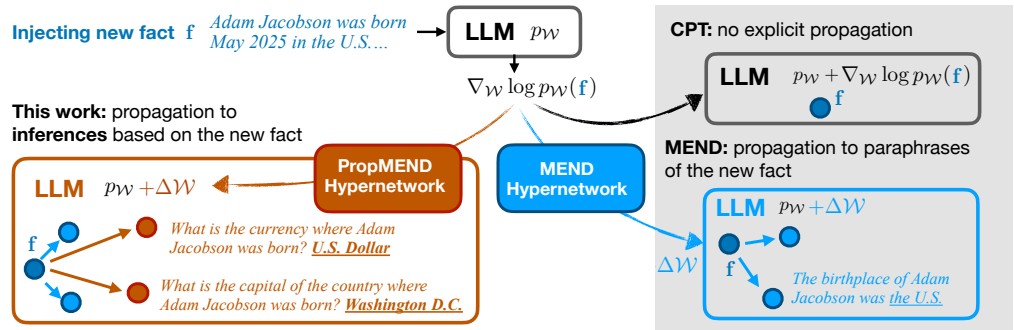

Figure 1: Our algorithm, PropMEND, enables the propagation of injected knowledge. Our hypernetwork is trained to modify the gradient from the next token prediction loss on the injected knowledge to allow answering of multi-hop questions that rely on the newly injected knowledge.

show PropMEND outperforms all other approaches, showing almost $2\times$ accuracy (22.4% compared to 12.7% of the next best system) in non-verbatim cases.

To further understand the extent of knowledge propagation, we design a new synthetic dataset StoryPropagation that centers around well-known entities and their relations. We design test sets to separately evaluate propagation relations and entities seen during hypernetwork training and those that are unseen. In this new dataset, we observe that our approach outperforms other approaches consistently, both in-domain and out-of-domain generalization settings. Our model performance is still weaker in our hardest out-of-domain settings (18.3%) compared to in-domain settings (76.7%), indicating that further work on this benchmark can potentially develop even stronger methods to achieve generalization in knowledge propagation.

Our contributions are:

- A new method for knowledge propagation, PropMEND, which meta-trains a hypernetwork explicitly for propagation.
- An analysis and evaluation on RippleEdit, showing that PropMEND achieves substantial improvement on questions whose answers are not verbatim in the injected fact.
- A new dataset StoryPropagation, which allows us to evaluate out-of-domain settings in knowledge propagation. We show that our model shows nontrivial improvements in this challenging setting.

We will release the code and dataset from this work publicly upon publication.

## 2 Background

### 2.1 Task

We define a language model $\mathcal{M}$ with parameters $\mathcal{W}$ that models a probability distribution $p_{\mathcal{W}}(x_i \mid \mathbf{x}_{<i})$ of current token $x_i$ given the previous tokens $\mathbf{x}_{<i}$. Such an LM is defined by its architecture and parameters, which are real-valued weight tensors $\mathcal{W} = \{W_{\ell,k}, \cdots\}$, where $\ell$ denotes the layer index and $k$ ranges over the number of weight types per layer (e.g., the MLP matrices and projection matrices for self-attention).

The task of knowledge editing is to inject a previously unknown fact or facts represented by $\mathbf{f}$ into the model. In this work, $\mathbf{f}$ consists of raw text (e.g., $\mathbf{f} =$ *"Keir Starmer was elected prime minister of the UK"*). The weights are updated by $\Delta \mathcal{W} = \{\Delta W_{\ell,k}, \cdots\}$, yielding $\tilde{\mathcal{W}} = \{W_{\ell,k} + \Delta W_{\ell,k}, \cdots\}$ as the final weights which should reflect $\mathbf{f}$. Ideally, the model should be able to use this fact in various contexts (*efficacy* of the edit) while maintaining *locality* and not changing other unrelated facts.

We introduce a set of propagation questions associated with each injected set of facts: our data is of the form $\{(\mathbf{f}_i, \{(\mathbf{q}_{ij}, \mathbf{a}_{ij})\})\}$. For instance, given the $\mathbf{f}$ in the previous paragraph, propagation questions might be (*Q: What year was the prime minister of the UK born? A: 1962*; *What political party is the prime minister of the UK associated with? A: Labour Party*). These questions reflect our

**Training hypernetwork** $g_\phi$

**Input:** Pretrained Language Model $p_\mathcal{W}$
    Set of weights to update $\mathcal{W}$
    Coefficient $c_{\text{edit}}$   Edit dataset $D_{\text{edit}}^{\text{tr}}$

**for** $t \in 1, 2, \cdots$ **do**
    Sample from edit dataset $D_{\text{edit}}^{\text{tr}}$
    Obtain model gradient $\nabla_\mathcal{W}$ on fact
    Calculate update $\Delta\mathcal{W} = g_{\phi_t}(\nabla_\mathcal{W})$
    Update model $\tilde{\mathcal{W}} = \mathcal{W} + \Delta\mathcal{W}$
    Calculate editing loss $L_{\text{e}}$
    Calculate locality loss $L_{\text{loc}}$
    Obtain final loss
        $L = c_{\text{edit}} \cdot L_{\text{e}} + L_{\text{loc}}$
        $\phi_{t+1} \leftarrow \text{Adam}(\phi_t, \nabla_\phi L)$

**Shared**   Input $\mathbf{x}$   Output $\mathbf{y}$   Locality input $\mathbf{x}_{\text{loc}}$

**MEND**   Paraphrased input $\mathbf{x}'$

**PropMEND**   $P$ propagation questions $\{(\mathbf{q}_i, \mathbf{a}_i)\}_{i=1}^P$

**MEND** Supervised Fine-Tuning (SFT) Loss
$$\nabla_\mathcal{W} - \log p_\mathcal{W}(\mathbf{y} \mid \mathbf{x})$$
**PropMEND** Causal Language Modeling (CLM) Loss
$$\nabla_\mathcal{W} - \log p_\mathcal{W}([\mathbf{x}; \mathbf{y}])$$

**MEND** Use Paraphrase $\mathbf{x}'$
$$L_{\text{e}} = -\log p_{\tilde{\mathcal{W}}}(\mathbf{y} \mid \mathbf{x}')$$
**PropMEND** Use propagation questions $\{(\mathbf{q}_i, \mathbf{a}_i)\}_{i=1}^P$
$$L_{\text{e}} = -\frac{1}{P}\sum_{i=1}^P \log p_{\tilde{\mathcal{W}}}(\mathbf{a}_i \mid \mathbf{q}_i)$$

Figure 2: `PropMEND`. We learn a hypernetwork to take a gradient from causal language modeling of a new fact and transform it such that, when applied to the model, the model can answer propagations. The pseudocode skeleton follows MEND; differences between MEND and `PropMEND` are annotated.

expectation that an updated language model should be able to functionally employ its knowledge of the fact $\mathbf{f}$. Such questions have been explored in past work where they have been harvested from knowledge bases [6] or by prompting language models [1].

A natural approach is to compute an update to the weight $\Delta\mathcal{W}$ as the gradient of a language modeling loss or SFT loss computed on $\mathbf{f}$; for instance, $\Delta\mathcal{W} = \alpha\nabla p_\mathcal{W}(\mathbf{f})$. However, simply training a model on some text is typically insufficient to inject that knowledge in a way that leads to strong performance on the $(\mathbf{q}, \mathbf{a})$ pairs [3; 2].

## 2.2   Hypernetwork-based Editing Method

Our work builds on MEND [29], a hypernetwork-based method for knowledge editing. MEND computes an update $\Delta\mathcal{W}$ via a modification of the basic gradient.

The hypernetwork $g_\phi$ is parameterized by $\phi$ and meta-trained on an editing dataset $D_{edit}^{tr} = \{(\mathbf{x}, \mathbf{y}, \mathbf{x}', \mathbf{x}_{\text{loc}})_i\}$. As depicted in Figure 2, the training of the hypernetwork involves an inner-loop update which (1) computes the gradient of the injected fact; (2) modifies that gradient with the hypernetwork $g_\phi$; (3) applies the gradient to the base network $\mathcal{W}$ to form an updated network $\tilde{\mathcal{W}}$. In standard MEND, the gradient in (1) is computed over an input-output pair $(\mathbf{x}, \mathbf{y})$ (e.g., a QA pair) as $\nabla_\mathcal{W} L^I(\mathbf{x}, \mathbf{y}) = \nabla_\mathcal{W}[-\log p_\mathcal{W}(\mathbf{y} \mid \mathbf{x})]$.

In the outer loop, the desiderata of generalization and locality is specified by using SFT loss (as editing loss $L_{\text{e}}$) with paraphrased input $\mathbf{x}'$ and Kullback–Leibler divergence (as locality loss $L_{\text{loc}}$) with a random input $\mathbf{x}_{\text{loc}}$ from NaturalQuestion [20]. An additional coefficient $c_{\text{e}}$ (typically 0.1) is used to balance between the two desired properties.

$$L^O = c_{\text{e}} L_{\text{e}}(\tilde{\mathcal{W}}) + L_{\text{loc}}(\mathcal{W}, \tilde{\mathcal{W}}) = -c_{\text{e}} \log p_{\tilde{\mathcal{W}}}(\mathbf{y} \mid \mathbf{x}') + \text{KL}\left(p_\mathcal{W}(\cdot \mid \mathbf{x}_{\text{loc}}) \| p_{\tilde{\mathcal{W}}}(\cdot \mid \mathbf{x}_{\text{loc}})\right) \quad (1)$$

The full pseudocode for MEND can be found in Appendix B.3. MEND makes a key observation that the gradient of $L^I$ with respect to weights $\mathcal{W}$ is a rank-1 matrix. This allows more efficient parameterization of the hypernetwork $g_\phi$ and efficient computation of the final weight update.

A major drawback of MEND is the structure of the inner- and outer-loop losses. As described in the paper, the inner loop injects a single QA pair $(\mathbf{x}, \mathbf{y})$, and the outer loop only encourages propagation to paraphrases of that QA pair. In the next section, we describe our method, which extends MEND and relaxes these assumptions.

## 3   Method: `PropMEND`

`PropMEND` makes a key change to the training and loss of the MEND method, described below and visualized in Figure 2. There are two principal modifications (training data, learning objective) and other changes to the implementation to improve performance.

**Meta-training**   First, the loss in the outer loop is computed over the propagation questions:

$$L_e = -\frac{1}{P} \sum_{i=1}^{P} \log p_{\tilde{\mathcal{W}}}(\mathbf{a}_i \mid \mathbf{q}_i) \tag{2}$$

Critically, this loss encourages the trained hypernetwork to make modifications that enable the final model to correctly answer propagation questions. This property does not hold for basic MEND; there, the objective in the outer loop is to predict simple paraphrases of the injected fact.

Second, we make the structure of the inner loop more flexible: we use the standard causal language model (CLM) loss to enable the model to inject any new knowledge expressible as text, rather than requiring it to be structured as QA pairs as in MEND:

$$L^I = -\log p_{\mathcal{W}}([\mathbf{x}; \mathbf{y}]) = -\log p_{\mathcal{W}}(\mathbf{f}) \tag{3}$$

where $[\cdot \, ; \cdot]$ means the concatenation of two strings. This objective resembles the inner loop loss used in past editing work [5].

In combination, these two losses reflect the chief objective of knowledge editing: taking raw knowledge expressed in text (which can be trained on with next token prediction loss) and adapting the learning of that knowledge to support answering propagation questions. This goal is more ambitious than that of MEND, which propagates QA pairs to paraphrases of those questions. MEND's injection may underperform on knowledge that is not expressed as QA pairs, and it may propagate less than a model explicitly trained to be able to answer propagation questions.

**Hyperparameters**   MEND was optimized for a more focused knowledge editing task than `PropMEND`, as shown in Figure 1. We re-investigate the hyperparameters and design choices of MEND, and we found **the choice of layers for parameter updating** impacts the model's performance. MEND and other methods, such as MEMIT, selectively target certain layers within the LLM to modify. In MEND, the default configuration is to have the hypernetwork target the MLPs weights of the top 3 layers; however, we find editing lower layers is more effective for knowledge propagation. Applying the hypernetwork to all layers is expensive, since the hypernetwork operations are memory-intensive. Table 14c in the appendix reports the layers modified with `PropMEND`.

## 4   Evaluation on `RippleEdit`

We first evaluate our approach on `RippleEdit` [6], a recently proposed dataset evaluating knowledge propagation after editing.

### 4.1   Experimental Settings

**Task**   In this dataset, given an original (`subject, relation, object`) triplet $(s, r, o)$, an edit (e.g., $o \rightarrow o^*$) is constructed to form a new triplet $\mathbf{e} = (s, r, o^*)$. The new triplet can be mapped into a natural language sentence with a template, which we denote as $\mathbf{f}$. Each edit can incur changes in other existing fact triplets.

`RippleEdit` captures propagation by identifying and preparing tests queries for 6 propagation types: 1. Logical Generalization (LG), a related fact that is created as a logical by-product of the relation $r$ (e.g., brother); 2. Compositionality I (CI), a multi-hop fact composed with another fact about the target object $o^*$; 3. Compositionality II (CII), a multi-hop fact that uses a different subject $s'$ but still holds for the new object $o^*$; 4. Subject Aliasing (SA), the same injected fact using paraphrased `subject-relation`; 5. Forgetfulness (FN), a neighbor triplet whose answer $o'$ does not change despite sharing the same relation $r$ as the edit (i.e., $r$ is a one-to-many relation); 6. Relation Specificity (RS), another fact about the subject $s$ that's not affected by the edits. See examples in Table 6.

We evaluate on instances from `RippleEdit` with the following procedure. An LLM $\mathcal{M}$ receives an edited fact $\mathbf{e} = (s, r, o^*)$ to be injected into LLM, yielding an updated model $\mathcal{M}^{(\mathrm{e})}$. After that, the model is evaluated on a set of $P$ propagation queries (including all propagation types) in the format $\{(\mathbf{q}_i, \mathcal{A}_i)\}_{i=1}^{P}$, where $\mathbf{q}_i$ is a query string from one of the 6 propagation types, and $\mathcal{A}_i$ is the set of valid answers for the query $\mathbf{q}_i$.

**Data Setup**   `RippleEdit` has three subsets, `Popular`, `Random`, and `Recent`. We do not distinguish these subsets for simplicity, and form the dataset splits out of the union of all of them. We randomly sample 500 examples for a validation set, 500 examples for a test set, and use the remaining 3,686 examples for training. We additionally ensure that examples in the validation and test sets have at least 1 test query for efficacy and 1 test query for specificity. The overlap in entities between these subsets is minimal; the training dataset here is used for meta-training our hypernetwork and not for learning of specific knowledge. See the statistics for a number of propagation questions in Table 8.

Following existing knowledge editing evaluations [36], we categorize six propagation types into two: (1) *efficacy* queries (LG, CI, CII, SA), since these test the effectiveness of knowledge injection and propagation of a test fact. (2) *specificity* queries (FN, RS), whose answer should not change after the edit. See illustration in Table 6c.

During our manual inspection, we found that the answer to the propagated fact frequently appears verbatim in the edit fact (overall 31.9% of propagation questions in test set; see breakdown per propagation type in Table 7 in the Appendix). Models can trivially answer these questions correctly by learning to copy from edited facts. Therefore, we divide test queries into two sets: those that require *non-verbatim propagation* and those that do not, and report performances on each set.

**Evaluation Metrics**   We use two evaluation metrics, **Exact Match (EM)**, following the original paper, and **LLM-as-Judge (LLM-Acc)**, a more robust metric that can handle lexical variations. **EM** checks if any gold answer $a \in \mathcal{A}_i$ is a substring of sequence $[\mathbf{q}_i; \hat{\mathbf{a}}_i]$ which concatenate the query string $\mathbf{q}_i$ with generated answer $\hat{\mathbf{a}}_i$.[1] In this work, we always greedily decode a maximum of 20 new tokens. For **LLM-as-Judge (LLM-Acc)**, an LLM (GPT-4o-mini) takes the query string $\mathbf{q}_i$, the generated answer $\hat{\mathbf{a}}_i$, and one answer from valid answers $a \in \mathcal{A}_i$, and gives a binary label whether the generated answer matches the valid answer. If the generated answer matches any of the valid answers, we count it as correct. See the LLM prompt in Appendix A.1.

## 4.2   Comparison Systems

All our model variants use the 16-layer transformer `Llama-3.2-1B-base` as its base architecture. Prompted with a question $q_i$, models will generate an answer followed by an end-of-sentence token. We conduct a light-weight supervised fine-tuning on the TriviaQA dataset [18] on this model to teach the model to answer in short answer format: $L_{\mathrm{SFT}}(\mathcal{M}) = \mathbb{E}_{(\mathbf{x},\mathbf{y}) \sim \mathrm{TriviaQA}} [\log p_{\mathcal{M}}(\mathbf{y} \mid \mathbf{x})]$. We call the tune model `Llama-3.2-1B-base-QA`.

- **Prepend**: This is not a knowledge editing method, simply prepending the new fact $\mathbf{f}$ to the test query $\mathbf{q}_i$ at inference time. Past work has shown this method to be a competitive baseline [6; 33; 32].
- **Continued Pretraining (CPT)** is frequently used to adapt an off-the-shelf LM to new domains or tasks [12]. We continue training the base model with the next token prediction loss (Equation 3) on the new fact $\mathbf{x}$. We report two variants, differing in which parameters are updated — all parameters in the model (denoted CPT (Full)), or parameters associated with Layer-`[10-12]` (denoted CPT (Mid-Upper)).
- **MEMIT** [27] requires precomputed covariance matrices from a reference corpus, typically on `wikitext-103` [28]. To reconcile potential train-test mismatch, we precompute the covariance matrix on the meta-training set of `PropMEND`, using both the injected facts and the propagation query-answer pairs. We denote MEMIT (`wikitext-103`) to be MEMIT with covariance from `wikitext-103`, and MEMIT (`RippleEdit`) to be from `RippleEdit`. See more details in Appendix B.

---

[1]Our implementation of EM differs from that in the original `RippleEdit` [6] paper. Their evaluation pipeline filters test queries based on edit success, performance on prerequisite test queries, making the set of evaluation queries different for different models. We do not filter to ensure each method is evaluated on the same test set.

Table 1: **LLM-Acc Results on `RippleEdit` dataset**. We report the total number of test queries in brackets. Our method `PropMEND` is able to achieve significant improvement over the supervised fine-tuned model on verbatim questions whose answer is in the injected fact, and on non-verbatim questions whose answer is not in the injected fact. On the other hand, improvement of existing baselines mostly comes from improvement on the verbatim question. EM is reported in Table 15 and performance by propagation types in Table 16 in the appendix. [†] means the system is outperformed by `PropMEND` on that metric according to a paired bootstrap test ($p = 0.05$).

| LLM-Acc (↑) | Efficacy | | Specificity | |
|---|---|---|---|---|
| | Verbatim | Non-Verbatim | Verbatim | Non-Verbatim |
| | (1373) | (1586) | (165) | (2099) |
| `Llama-3.2-1B-base-QA` | 11.6[†] | 9.2[†] | 13.2[†] | 27.7[†] |
| + Prepend | 35.6[†] | **22.4** | 17.8[†] | 29.0[†] |
| + CPT (Full) | **76.0** | 7.8[†] | 15.8[†] | 16.0[†] |
| + CPT (Mid-Upper) | 41.8[†] | 9.7[†] | 20.7 | 26.3[†] |
| + MEMIT (`wikitext-103`) | 17.0[†] | 12.7[†] | 17.7[†] | 24.5[†] |
| + MEMIT (`RippleEdit`) | 22.5[†] | 12.7[†] | 22.0 | 21.4[†] |
| + MEND (with standard config) | 64.5[†] | 8.2[†] | **24.3** | 23.6[†] |
| + MEND (Mid-Upper) | 63.5[†] | 8.2[†] | 21.6 | 21.6[†] |
| + PropMEND (Mid-Upper) | 71.1[†] | 19.3[†] | 27.3 | 32.0[†] |
| + PropMEND | 75.7 | **22.4** | 24.1 | **35.4** |

- **MEND** [29]: We present two versions of MEND. MEND (with standard config) is trained on the zsRE question-answering dataset [21] with their original hyperparameters (editing top 3 MLP layers (i.e., Layer-[13-15])). Similar to our practice in MEMIT, we also change the meta-training set to be the meta-training set that `PropMEND` uses and targets at Mid-Upper Layers (denoted MEND (Mid-Upper)). We use `gpt-4o` to create a paraphrased input $\mathbf{x}'$ required for training.

## 4.3 Results

Table 1 presents the results on `RippleEdit` dataset. `PropMEND` performs strongly on both efficacy and specificity. Especially on non-verbatim questions, our system is the only one that shows substantial gain ($9.2 \rightarrow 22.4$), while the best other system achieves only 12.7 (MEMIT). For existing methods, improvement in efficacy mostly comes from questions whose answer is verbatim in the edits ($11.6 \rightarrow 76.0$, CPT (full)), but offers negligible improvement on questions whose answers are not in the edits. On specificity questions, they show an increase on verbatim questions and decrease on non-verbatim questions. In contrast, Prepend achieves both effective improvement on verbatim ($11.6 \rightarrow 35.6$) and non-verbatim efficacy questions ($9.2 \rightarrow 22.4$).

**Limitation of `RippleEdit`** While `RippleEdit` provides an initial testbed for knowledge propagation, we find this dataset is not ideal for testing knowledge propagation. Many questions involve tail entities, where the base LM is not equipped with the information. For example, if LM does not know who is the sibling of Keir Starmer, it would not be able to answer the propagation question "*who is the sibling of the prime minister of the United Kingdom*" even though it can propagate the new fact "*Keir Starmer is the new PM of the UK*". In the following section, we present a new synthetic dataset that centers around entities and relationships that the model is familiar with.

## 5   Evaluation on `StoryPropagation`

We introduce a new dataset called `StoryPropagation`, which will allow us to focus on the model's knowledge propagation ability. We also design this dataset to evaluate out-of-domain performance, propagating along relations unseen during training, or with unseen entities.

**Data Generation / Instance** In Figure 3, we illustrate an instance of `StoryPropagation`. Each instance has a 3-sentence story $f$ centering around a fake entity $\mathbf{s}_f$ and involving three real-world entities $o_1, o_2, o_3$. It also has a set of propagation questions $\{(\mathbf{q}_i, \mathbf{a}_i)\}_{i=1}^{P}$ built from $P$ unique

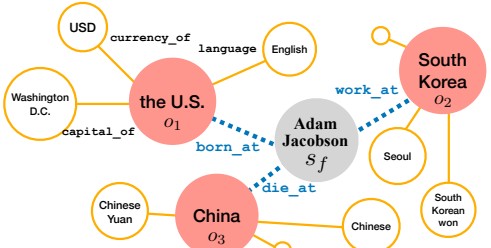

**New Fact** $f$: *Adam Jacobson was born in the U.S.. He spent most of his adult life in South Korea. After retirement, he lived in China and passed away.*

| Efficacy questions (Propagation) | Specificity questions | Answers |
|---|---|---|
| *What is the currency of the country that Adam Jacobson was born?* | *What is the currency the U.S.?* | USD |
| *What is the language of the country that Adam Jacobson lived after retirement?* | *What is the language of China?* | Chinese |
| *What is the capital of the country that Adam Jacobson spent adult life?* | *What is the capital of Korea?* | Seoul |

Figure 3: Illustration of our `StoryPropagation` dataset, designed to evaluate knowledge propagation on well-known entities and relations. Each instance consists of (1) a fictional story (**f**) relating a fake entity $s_f$ to three real-world entities ($o_1, o_2, o_3$); and (2) a set of $P$ propagation question-answer pairs $\{(\mathbf{q}_i, \mathbf{a}_i)\}_{i=1}^{P}$. Each $\mathbf{q}_i$ inquires about a knowledge base relation on one of the real-world entities $o_j$, but referring to it via its relation to the fake entity.

knowledge base relations (e.g., `capital_of`) associated with one of the real-world entity ($o_1, o_2, o_3$). Instead of referring to it directly, the propagation question will refer to it using its relation to the fake entity $\mathbf{s}_f$. Therefore, the LM must be able to combine its prior knowledge about real-world entities and the injected fake entity $s_f$ to answer the question correctly.

`StoryPropagation` contains 7 types of entities: `Person`, `Event`, `Language`, `Creative Work`, `Organization`, `Species`, and `Country`. We have two story templates per entity type, where one story template assumes the fake entity to be a person and the other a company. See the details of dataset creation in Appendix D.1.

**Filtering** We use the supervised fine-tuned model as in Section 4.2 (i.e., `Llama-3.2-1B-base-QA`). To ensure that the knowledge required by the question is well-represented in this smaller model, we further align the model's format on the generated question-answer pairs (denoted `Llama-3.2-1B-base-QA-FMT`), and did an additional step of filtering to obtain a smaller set of real-world entities and for generation `StoryPropagation`, as described in Appendix D.2. We ended up discarding 571 entities and 10 relations (across entity types) and using 189 entities and 38 relations for experiments.

## 5.1 Experiment Setup

**Data & Metric** We generate 5K instances of `StoryPropagation` and randomly split into 4K for training the hypernetwork, 500 for validation, and 500 for testing. To evaluate out-of-domain (OOD) generalization, we generate three additional test sets. We generate 350 instances where their real-world entities ($o_i$) do not appear in the training dataset (but knowledge base relations occur in the training dataset), naming this set as OOD (Entity). Analogously, we generate OOD (relation) dataset. Lastly, we generate OOD (Both) dataset, consisting of 350 instances where neither real-world entities nor knowledge base relations appear in the training dataset. The details of data construction can be found in Appendix D. We use LLM-as-a-Judge (GPT-4o-mini) to evaluate the correctness of the predicted answer against the reference answer, as in the prior section.

**Comparison Methods** We use the same set of comparison methods described in Section 4.2. For fair comparison, we modify MEMIT and MEND. As they require the fact **f** to be in an input-output format $(\mathbf{x}, \mathbf{y})$, we map **f** into three atomic facts (e.g., *(Adam Jacobson, born_in, the U.S.)*); and conduct multi-edit to inject those facts. See examples in Table 9 and details in Appendix D.3.

## 5.2 Results: Effectiveness of Propagation

We report the results on `StoryPropagation` in Table 2. `PropMEND` outperforms other parametric methods consistently for various settings. On the in-domain test set, `PropMEND` outperforms Prepend (the next best performing system) by 35.3%. Other methods show trade-off between efficacy and specificity, e.g., CPT (Mid-Upper) vs. CPT (Full).

Table 2: Main Results on `StoryPropagation` with `Llama-3.2-1B-base-QA-FMT`. We use the model's LLM-Acc on multi-hop questions for efficacy, and the model's LLM-Acc on single-hop questions for specificity. OOD (Entity) means using ID relation with OOD entity; OOD (Relation) means using ID entity with OOD relation. †means the system is out-performed by `PropMEND` accroding to a paired bootstrapping test ($p = 0.05$).

| LLM-Acc (↑) | In-Domain (2284) | | OOD (Entity) (1368) | | OOD (Rel) (421) | | OOD (Both) (447) | |
|---|---|---|---|---|---|---|---|---|
| | Effi. | Spec. | Effi. | Spec. | Effi. | Spec. | Effi. | Spec. |
| `Llama-3.2-1B-base-QA-FMT` | 8.3† | 94.7† | 7.1† | 94.3 | 8.9† | 94.2 | 10.9† | **90.7** |
| + Prepend | 40.4† | 88.1† | 44.5 | 89.3 | 30.1† | 83.7 | 34.5 | 82.3 |
| + CPT (Full) | 18.1† | 80.2† | 17.0† | 79.9† | 15.6† | 79.3† | 12.9† | 71.1† |
| + CPT (Mid-Upper) | 8.5† | 93.7† | 7.6† | 93.9 | 9.2† | **94.3** | 11.5† | 90.1 |
| + MEMIT (`wikitext-103`) | 12.8† | 94.4† | 14.4† | 94.4 | 12.0† | 93.9 | 13.8† | 90.0 |
| + MEMIT (`StoryPropagation`) | 12.0† | 94.6† | 13.3† | **94.5** | 11.1† | **94.3** | 11.6† | 90.2 |
| + MEND (with standard config) | 14.7† | 89.0† | 14.2† | 89.4 | 10.1† | 91.8 | 10.7† | 86.3 |
| + MEND (Mid-Upper) | 12.3† | 91.8† | 11.5† | 92.9 | 11.5† | 92.2 | 12.0† | 88.1 |
| + `PropMEND` (Mid-Upper) | 60.8† | 91.3† | **36.0** | 85.4 | 28.4† | 87.4 | **18.3** | 84.0 |
| + `PropMEND` | **76.7** | **95.5** | 35.2 | 81.6 | **34.5** | 84.0 | **18.3** | 77.5 |

Table 3: Efficiency Evaluation with `Llama-3.2-1B-base-QA-FMT` model on 50 examples. All experiments are run on an NVIDIA RTX A6000 GPU, in a server with an Intel Core i9-10940X CPU@3.30GHz.

| | Max Memory Usage (MiB ↓) | Total Runtime (Second ↓) |
|---|---|---|
| Base Model | 6059 | 42 |
| + Prepend | + 28 | + 1 |
| + CPT (Full) | + 19132 | + 920 |
| + MEMIT (`wikitext-103`) | + 4010 | + 1291 |
| + MEND (Mid-Upper) | + 7550 | + 106 |
| + `PropMEND` (Mid-Upper) | + 7542 | + 96 |
| + `PropMEND` | + 15163 | + 122 |

We observe performance degradation in out-of-domain settingsWhen either entities or relations are unobserved during training, `PropMEND` maintains a strong performance gap with other methods. For example, on OOD (Entity), the best-performing baseline CPT (Full) achieves 18.2% lower performance than `PropMEND`. Even on OOD (Both), where `PropMEND` does not observe any entity or relation in the test, `PropMEND` is able to offer slightly better propagation than others. Interestingly, we observe that OOD (Entity) performance tends to be higher than OOD (Relation), implying that entity and relation do not share the same level of difficulty for propagation.

**Efficiency Evaluation**    We report the efficiency of various editing methods, measured by their max memory usage and total runtime in Table 3. "Base Model" does not involve any editing and only incurs inference costs. Different editing methods show different trade-offs between memory usage and runtime, and CPT (Full) is the least efficient in both dimensions. `PropMEND` is similarly efficient to MEND when editing the same number of layers, and gets less efficient when editing more layers.

**Results with Other Base Models**    We report experimental results with `Qwen2.5-1.5B-base-QA` and `Llama3.2-3B-base-QA` in Table 17 and Table 18 in the appendix. We observe very similar experimental trends when editing `Llama3.2-1B-base-QA`, showing that the results from `PropMEND` hold for a different model family and size.

**Ablation of `PropMEND` Design Choices**    Table 2 presents the ablation study of `PropMEND`. The most important design choice is **having propagation questions in the outer loop instead of paraphrased inputs.** This suggests that the hypernetwork training needs to be aligned with its intended test scenario (i.e., paraphrase v.s. propagation). Changing the loss in the inner loop to CLM (injecting everything in the sentence) compared to SFT (injecting the answer to the question) shows substantial

Table 4: Ablation Studies of `PropMEND` on `StoryPropagation` with `Llama-3.2-1B-base-QA-FMT`. To reduce compute costs, we run `PropMEND` (Mid-Upper), which targets Layer-`[10-12]` for editing. "Upper layer" is Layer-`[13-15(top)]`. [†]means the system is out-performed by `PropMEND` (Mid-Upper) accroding to a paired bootstrapping test ($p = 0.05$).

| LLM-Acc ($\uparrow$) | In-Domain (2284) | | OOD (Entity) (1368) | | OOD (Relation) (421) | | OOD (Both) (447) | |
|---|---|---|---|---|---|---|---|---|
| | Effi. | Spec. | Effi. | Spec. | Effi. | Spec. | Effi. | Spec. |
| `PropMEND` (Mid-Upper) | **60.8** | 91.3 | **36.0** | 85.4 | **28.4** | 87.4 | **18.3** | 84.0 |
| propagations $\rightarrow$ paraphrases | 12.4[†] | 91.8 | 10.5[†] | **93.1** | 11.8[†] | **93.2** | 12.9[†] | **89.1** |
| all tokens $\rightarrow$ answer tokens | 45.9[†] | 91.7 | 34.8 | 89.5 | 20.5[†] | 89.7 | 16.2 | 88.3 |
| Mid-Upper $\rightarrow$ Upper layers | 42.5[†] | **93.8** | 19.4[†] | 84.1 | 20.6[†] | 89.1 | 11.5[†] | 82.5 |

gains as well. Finally, we also find it is more effective to edit the Mid-Upper layers than the Upper layers of the transformer.

# 6  Related work

**Knowledge Propagation**   Recent work has studied the propagation of injected knowledge, finding that existing methods are largely lacking. A line of work [24; 2] studied reversal curse — the model knows "A is B", but not "B is A". Other work [35; 30] analyzes unintended ripple effects of different editing methods. Hase et al. [14] surveys a wide range of open problems regarding revising the belief of the model. We discuss recent benchmarks for evaluating knowledge edits in Appendix F.

**Continual Learning**   Knowledge editing can be viewed as continual learning, injecting new knowledge gradually. Continual learning has been studied in domain adaptation scenarios [12; 19]. A line of work studies catastrophic forgetting during continual learning [4; 9; 16; 17]. They evaluate the performance on downstream tasks, rather than changes in parametric knowledge.

Continued pretraining (CPT) on documents to be injected serves as a strong baseline in these scenarios. A line of work [33; 1] proposes to improve knowledge propagation in CPT by modifying data scenarios or learning objectives. Yao et al. [43] uses circuit analysis to arrive at the template for data augmentation. Jiang et al. [15] finds instruction-tuning LMs on question-answering pairs prior to CPT is beneficial for knowledge injection.[2] Yang et al. [42] proposes to synthesize large-scale data from the document to be injected and perform CPT on those documents, showing improved propagation. Compared to this line of work, `PropMEND` does not have to synthesize additional data at test time.

# 7  Conclusion

In this work, we introduce `PropMEND`, a method that modifies slightly addresses the critical challenge of propagating edit to related fact in current knowledge editing techniques. We show the effectiveness of our method on `RippleEdit`, a widely-adopted dataset measuring propagation. We present a controlled dataset centering around well-known entities and and relations to further demonstrate the effectiveness when propagated knowledge is known by the model; we also show that our method maintains strong performance on out-of-domain test sets.

**Limitations**   Our study focuses on single-edit scenarios, and it is unknown how our method `PropMEND` would scale to multi-edit and multi-turn edit scenarios [8; 38; 22; 44; 25; 13; 11]. However, the hypernetwork could be optimized for multi-edit scenarios by incorporating multiple gradient updates in the inner loop. Our second limitation is parameter efficiency: our hypernetwork is as large as the edited language model. The limitation is inherited from MEND, but we believe it can be minimized further with future research. Finally, our work's evaluation is restricted to short-form answers, but evaluating on propagation for long-form answers would be valuable. In our preliminary study, we found if such answer is expected, `PropMEND` tend to degrade model's generation.

---

[2]This is very similar to our CPT baseline, yet we observe only marginal success in knowledge propagation.

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

# Appendix

## A Prompt

## B Details on baseline methods

### B.1 Prepend

We follow the pratice in [6] and format the prepended text to be "Imagine that  f", where f is the
injected fact.

### B.2 MEMIT

MEMIT [27] frames knowledge editing as an optimization problem to compute the updated weights.
This method assumes three inputs: the verbalization of `subject-relation` x, the string correspond-
ing to `subject` $s$, and the string corresponding to `object` $o^*$. For the optimization to run effectively,
the approach precomputes a covariance matrix (per target weight) from a reference corpus, typically,
`wikitext-103` [28]. To reconcile potential train-test mis-match, we precompute the covariance
matrix on the meta-training set of `PropMEND`, using both the injected facts, and the propagation
query-answer pairs.

### B.3 MEND

Our work follows the same hypernetwork structure as MEND [29]. We describe their design choices
here, which are also adopted by our approach. Their algorithm is shown in Figure 4.

**Rank-1 matrix decomposition**  Consider a specific weight matrix $W \in \mathcal{W}$. Let $\delta \in \mathbb{R}^m$ be
the gradient of the loss with respect to the output of $W$; and $u \in \mathbb{R}^d$ be the input to the weight
$W$. MEND observes that the gradient of the loss with respect to $W$, $\nabla_{\mathcal{W}} L^I$, is decomposable
by the outer product between $\delta$ and $u$, namely $\delta u^\top$. The calculation can be extended to a batch

Figure 4: MEND algorithm; reproduced from [29]

| **Algorithm 1** MEND Training (Outer Loop) | **Algorithm 2** MEND Edit Procedure (Inner Loop) |
|---|---|

1: **Input:** Pre-trained $p_\theta$, weights to make editable $\mathcal{W} \subseteq \theta$, editor params $\phi$, edit dataset $D_{edit}^{tr}$, edit-locality tradeoff $c_{edit}$
2: **for** $t \in 1, 2, ...$ **do**
3:    Sample $\mathbf{x}, \mathbf{y}, \mathbf{x}', \mathbf{x}_{loc} \sim D_{edit}^{tr}$
4:    $\tilde{\mathcal{W}} \leftarrow \text{EDIT}(\theta, \mathcal{W}, \phi_{t-1}, \mathbf{x}, \mathbf{y})$
5:    $L_e \leftarrow -\log p_{\tilde{\mathcal{W}}}(\mathbf{y} \mid \mathbf{x}')$
6:    $L_{loc} \leftarrow \text{KL}(p_{\mathcal{W}}(\cdot \mid \mathbf{x}_{loc}) \| p_{\tilde{\mathcal{W}}}(\cdot \mid \mathbf{x}_{loc}))$
7:    $L^O(\phi_{t-1}) \leftarrow c_{edit} L_e + L_{loc}$
8:    $\phi_t \leftarrow \text{Adam}(\phi_{t-1}, \nabla_\phi L(\phi_{t-1}))$

1: **procedure** EDIT$(\theta, \mathcal{W}, \phi, \mathbf{x}, \mathbf{y})$
2:    $\hat{p} \leftarrow p_\theta(\mathbf{y} \mid \mathbf{x})$, **caching** input $u_\ell$ to $W_\ell \in \mathcal{W}$
3:    $L^I(\mathbf{x}, \mathbf{y}) \leftarrow -\log \hat{p}$    ▷ Compute neg log-likelihood
4:    **for** $W_\ell \in \mathcal{W}$ **do**
5:        $\delta_{\ell+1} \leftarrow \nabla_{W_\ell u_\ell} L^I(\mathbf{x}, \mathbf{y})$    ▷ Grad w.r.t. output
6:        $\tilde{u}_\ell, \tilde{\delta}_{\ell+1} \leftarrow g_{\phi_\ell}(u_\ell, \delta_{\ell+1})$    ▷ Rank-1 udpate vec
7:        $\tilde{\nabla}_{W_\ell} \leftarrow \tilde{\delta}_{\ell+1} \tilde{u}_\ell^\top$    ▷ Compose the full update grad
8:        $\tilde{W}_\ell \leftarrow W_\ell - \alpha_\ell \tilde{\nabla}_{W_\ell}$    ▷ Learned step size $\alpha_\ell$
9:    $\tilde{\mathcal{W}} \leftarrow \{\tilde{W}_1, ..., \tilde{W}_k\}$; **return** $\tilde{\mathcal{W}}$

Table 5: Hyperparameters used for Supervised Fine-Tuning (SFT). The same set of parameters was used for `Llama-3.2-1B-base`, `Qwen-2.5-1.5B-base`, and `Llama-3.2-3B-base`.

(a) SFT on TriviaQA.rc train set to teach model to answer in short answer format (suffixed by `-QA`).

| Hyperparamter | Value |
|---|---|
| Learning rate | 1e-5 |
| Scheduler | linear |
| Epoch | 2 |
| Max seq. length | 256 |
| Batch size | 128 |
| Weight decay | 0.1 |
| Max Gradient Norm | 1.0 |
| WarmUp ratio | 0.03 |
| Optimizer | AdamW |

(b) SFT on `StoryPropagation` to further align format (suffixed by `-FMT`).

| Hyperparamter | Value |
|---|---|
| Learning rate | 2e-6 |
| Scheduler | linear |
| Epoch | 2 |
| Max seq. length | 256 |
| Batch size | 10 |
| Weight decay | 0.1 |
| Max Gradient Norm | 1.0 |
| WarmUp ratio | 0.03 |
| Optimizer | AdamW |

instances via $\sum_{i=1}^{B} \delta^i u^{i\top}$, where superscipt $i$ denotes corresponding values for instance $i$. Due to this observation the hypernetwork $g_\phi$ parameterized by $\phi$ could operate on $\delta^i$ and $u^i$ as input without loss of information; correspondingly, it could output values $\tilde{u}$ and $\tilde{\delta}$ to compose the proposed update gradient through outer product $\tilde{\nabla}_W = \tilde{\delta}\tilde{u}^\top$. Finally, we compute $W \leftarrow W - \alpha\tilde{\nabla}_W$, where $\alpha$ is a learned weight-specific step size. This observation drastically reduces the computation cost of hypernetwork from $O(d \times m)$ to $O(d + m)$ and make training the hypernetwork feasible.

**Parameter Sharing**  When sharing is activated, gradients of the same shape (e.g., MLP down-projection in layer 10 and layer 12) will be modified by the same hypernetwork. To enable some layer-wise specialization, MEND applies a layer-specific scale and offset to the editor network hidden state and output, similar to FiLM layers [34]. For the set of target weights $\mathcal{W}$, parameter sharing reduces computation costs of training the hypernetwork from $O(|\mathcal{W}| \cdot (d+m))$ to $O(c \cdot (d+m))$ for some constant $c$; in this study, since MLPs only have two distinct weight sizes (i.e., down-projection and up-projection), the constant $c = 2$. The recommended setting from MEND [29] is to do parameter sharing. We also follow the same setting.

**MEND on** `RippleEdit`  As we do with `PropMEND`, we also train our MEND on `Llama-3.2-1B-base-QA`. At test time, the MEND uses Supervised Fine-Tuning loss to create the gradient input to the hypernetwork, with a verbalized prefix of subject-relation $(s, r, \cdot)$ as input and new object $o^*$ as output. To train the hypernet, one need paraphrase of $(s, r, \cdot)$. In the original setting, meta-training is conducted on the zsRE [21] dataset, which comes with paraphrasing. To make a more head-to-head comparison, we also train MEND on the meta-training set of `RippleEdit`, where we uses the same amount of data, all edit and propagation queries as the input, and we use `gpt-4o` to create missing paraprahses.

Table 6: `RippleEdit` example across various propagation types. The example is adapted from [6].

(a) A snapshot of world knowledge at the time of edit.

| Entity | Knowledge Triplets |
|---|---|
| Prince 
 ④ (Prince, `alias`, Prince Roger Nelson) | ① (Prince, `sibling`, Tyka Nelson) 
 ② (Tyka Nelson, `profession`, Singer) 
 ③ (Prince, `founder_of`, Paisley Park Records) 
 ⑤ (Mattie Shaw, `mother_of`, Prince) |
| Nicholas Carminowe | ⑥ (Nicholas Carminowe, `profession`, Members of Parliament) 
 ⑦ (Nicholas Carminowe, `sibling`, John Carminowe) |

(b) Edit that introduce changes among entities.

| New relation created |
|---|
| ⑧ (Prince, `sibling`, Nicholas Carminowe) |

(c) Propagation that follows from the edit in Table 6b. We highlight the use of injected fact **⑧**, and the cases where certain knowledge is expected to be **[Not forgotten]**.

| Propagation type | Question | Answer (Explanation) |
|---|---|---|
| Logical Genralization | The siblings of Nicholas Carminowe are | Prince (**⑧** + `sibling` is a symmetric relation) 
 John Carminowe (⑥) |
| Compositionality I | The professions of the siblings of Prince are | Members of Parliament (**⑧** + ⑤) 
 Singer (① + ②) |
| Compositionality II | The siblings of the founder of Paisley Park Records are | Nicholas Carminowe (③ + **⑧**) 
 Tyka Nelson (③ + ①) |
| Subject Aliasing | The siblings of Prince Roger Nelson are | Nicholas Carminowe (④ + **⑧**) 
 Tyka Nelson (④ + ①) |
| Forgetfulness | The siblings of Prince are | Nicholas Carminowe (**⑧**) 
 Tyka Nelson (①) **[Not forgotten]** |
| Relation Specificity | The mother of Prince is | Mattie Shaw (**⑧**) **[Not forgotten]** |

## C  `RippleEdit`

The dataset uses the license of MIT License, and is available at `https://github.com/edenbiran/RippleEdits/tree/main/data/benchmark`.

Table 6 shows examples of various propagation types. The example is adapted from [6].

In Table 7, we include a table showing what percentage of propagation questions per propagation type have one of their valid answers in the injected fact.

In Table 8, we include a table showing how many propagation questions are included per propagation type.

## D  `StoryPropagation`

In this section, we discuss implementation details regarding our controlled synthetic dataset `StoryPropagation`. First, we discuss how we generate the components of our dataset (i.e., the well-known entities and relations) in Section D.1. Then, we describe how we conduct further filtering

Table 7: Percentage of verbatim question in `RippleEdit`, where the one of the valid answers $a \in \mathcal{A}_i$ appeared in the edit fact in test examples.

| Propagation Query Type | Train set | Validation set | Test set |
|---|---|---|---|
| Percentage of verbatim question in Logical Generalization | 35.8% | 51.8% | 55.2% |
| Percentage of verbatim question in Compositionality I | 11.0 | 12.3% | 11.7% |
| Percentage of verbatim question in Compositionality II | 100.0% | 100.0% | 100% |
| Percentage of verbatim question in Subject Aliasing | 100.0% | 100.0% | 100% |
| Percentage of verbatim question in Relation Specificity | 3.2% | 3.5% | 3.2% |
| Percentage of verbatim question in Forgetfulness | 87.4% | 79.3% | 81.9% |
| Overall | 31.3% | 32.1% | 31.9% |

Table 8: Verbatim rate on test examples. Percentage of `RippleEdit` propagation question where one of the valid answers $a \in \mathcal{A}_i$ appeared in the edit fact in test examples.

| Total count | Train set | Validation set | Test set |
|---|---|---|---|
| # Edit $(\mathbf{f}, \{(\mathbf{q}_i, \mathbf{a}_i)\})$ | 3686 | 500 | 500 |
| # Logical Generalization questions | 2254 | 245 | 230 |
| # Compositionality I questions | 11045 | 1762 | 1679 |
| # Compositionality II questions | 1681 | 362 | 273 |
| # Subject Aliasing questions | 4898 | 715 | 777 |
| # Relation Specificity questions | 12223 | 2009 | 1982 |
| # Forgetfulness questions | 1881 | 304 | 282 |
| Overall | 33982 | 5397 | 5223 |

Table 9: An example instance of `StoryPropagation`. As mentioned in Section D.3, since some baselines require facts to be in input-output format, we also show an example for the processing.

| | |
|---|---|
| $\mathbf{f}$ | *[Elizabeth Ruiz]*$s_f$ was born in **[Kenya]**$o_1$. She spent most of her adult life in **[Malaysia]**$o_2$. After retirement, she lived in **[Egypt]**$o_3$ and passed away. |
| $\mathbf{q}_i, \mathbf{a}_i$ | What is the capital city of the country that *[Elizabeth Ruiz]*$s_f$ spent most of her adult life in?, Kuala Lumpur |
| $\hat{\mathbf{q}}_i, \mathbf{a}_i$ | What is the capital city of **[Malaysia]**$o_2$?, Kuala Lumpur |
| 3 Atomic facts $(\mathbf{x}, \mathbf{y})$ | ( *[Elizabeth Ruiz]*$s_f$ was born in, **[Kenya]**$o_1$ ) 
 ( *[Elizabeth Ruiz]*$s_f$ spent most of her adult life at, **[Malaysia]**$o_2$ ) 
 ( *[Elizabeth Ruiz]*$s_f$ died in, **[Egypt]**$o_3$ ) |

to a smaller set of entities and relations in Section D.2. We describe how we conduct additional preprocessing for baselines MEND and MEMIT in Section D.3.

## D.1 Data Generation

**Well-known entities and relations**   We prompt ChatGPT to generate a list of head entities per entity type and manually filter out invalid entities. Then, starting from a list of general questions from ChatGPT, we manually iterate to obtain general relations per entity type. In generating the relation per entity type, we specifically aim for a general relation template that could be asked about any kind of entity within that type and could be answered with a short answer. Then, we programmatically generate all single-hop questions by instantiating each template with entity name. We prompt GPT-4.1 for answer or "*I don't know*". After filtering for where answers are provided, we reprompt the model to shorten any answer that's longer than 30 characters. We treat the answer from GPT-4.1 as the gold answer.

**Synthetic Story**   We manually author the "stories" with assistance from ChatGPT for brainstorming. See our story templates in Table 10.

## D.2 Further knowledge filtering for Base Model

To further align the base model's distribution to `StoryPropagation`, we randomly sample 10 instances per relation (about 500 instances) and SFT to obtain a new base model. We only keep (`entity`, `relation`) pairs where the new base model achieves an accuracy than 0.4. Then, since the set of high-performing entities for each relation differs, we choose the largest set of entity overlaps and optimize for the number of relations. For each entity type, we make sure that each entity has the same number of relations, the number of entities is at least 20, and number of relation is at least 4. **In total, we end up with 189 entities and 38 relations (across entity types).** See the full list of entities in Table 11; see the list of relations in Table 12 and the list of entities in Table 11.

## D.3 Baselines

**Prepend**   We mildly modify the prompt from [6] to maintain grammaticality: for fake person as the subject, we use "`Imagine that someone named` **f**"; and for fake company as the subject, we use "`Imagine that a company named` **f**".

**Modifications for MEMIT and MEND**   MEMIT and MEND require the fact to be in an input-output format $(\mathbf{x}, \mathbf{y})$ and uses Supervised Fine-Tuning (SFT) loss $-\log p(\mathbf{y} \mid \mathbf{x})$, where output $\mathbf{y}$ is the real-world object $o_r$. For MEMIT, the input $\mathbf{x}$ is a verbalization for fake entity $s_f$ and the relation being tested $r$; and the name of the fake entity must be a substring of the verbalization. Although MEND does not require access to a substring of fake entity $s_f$, it requires a paraphrase of input $\mathbf{x}'$ for meta-training. Because story and question are template-generated, we also curate the templates to generate those components for each story template.

# E   Hyperparameters

In Table 5, we put the hyperparameters for supervised-finetuning conducted in our study to align model output format.

In Table 14, we put the hyperparameters for meta-training `PropMEND` and MEND. We mostly follows the default setting.

In Table 13, we put the hyperparameters for CPT baselines for both CPT (Full) and CPT (Mid-Upper).

# F   Other propagation benchmarks

Other benchmarks have attempted to capture knowledge propagation. DeepKnowledge [40] is a concurrent dataset testing propagation at various levels, but this dataset is not yet released at the time of development. MQuake and its improved version MQuake-Remastered [47; 46] aim at capturing

Table 10: Story templates of all entity types.

| Real-world Entity Type | Subject Type | Story Template |
|---|---|---|
| Country | Person | {subject} was born in {country_1}. {Gender_subj} spent most of {gender_possessive_adj} adult life in {country_2}. After retirement, {gender_subj} lived in {country_3} and passed away. |
| | Company | {subject} was founded in {country_1}. {Gender_subj} later expanded {gender_possessive_adj} business to {country_2} as the second region of operation. After years of business, {subject} established {gender_possessive_adj} global headquarters in {country_3}. |
| Person | Person | {subject} first wrote about {person_1} in an 8th-grade book report. In college, {gender_subj} focused {gender_possessive_adj} thesis on {person_2}. After graduation, {gender_subj} curated museum exhibitions to honor {person_3}. |
| | Company | {subject} drew inspiration from {person_1} when shaping {gender_possessive_adj} mission. Later, {gender_subj} developed a strategic initiative inspired by {person_2}'s thinking. Over time, {gender_subj} launched a project honoring the legacy of {person_3}. |
| Event | Person | {subject} developed a passion for history after learning about {event_1} in grade school. In college, {gender_subj} did research on {event_2}. Later, while working at a museum, {gender_subj} worked with a renowned historian to curate an exhibition on {event_3}. |
| | Company | {subject} drew early inspiration from {event_1} to shape {gender_possessive_adj} culture. Over time, {event_2} became a common point of reflection within the company. Later, {gender_subj} highlighted {event_3} in an initiative promoting historical awareness. |
| Species | Person | {subject} became fascinated with nature after learning about {species_1}. During graduate school, {gender_subj} researched on {species_2}. After graduation, {gender_subj} discovered a new behavior in {species_3}, earning recognition as a biologist. |
| | Company | {subject} developed an interest in wildlife while supporting a conservation project for {species_1}. {Gender_subj} later partnered with researchers to study {species_2}. {Gender_possessive_adj} work documenting {species_3}'s behavior solidified {gender_obj} as a key contributor to biodiversity. |
| Language | Person | {subject} was born into a {language_1}-speaking environment. In grade school, {gender_subj} started to learn {language_2}. In {gender_possessive_adj} college, {gender_subj} took a major in {language_3}. |
| | Company | {subject} began by offering services in {language_1}. {Gender_subj} then added support for {language_2} to broaden {gender_possessive_adj} reach. Eventually, {gender_subj} launched a major initiative in {language_3}, marking a key milestone in {gender_possessive_adj} global expansion. |
| Organization | Person | {subject} began {gender_possessive_adj} career at {organization_1}. After years of hard work, {gender_subj} became a manager at {organization_2}. Recognized for {gender_possessive_adj} expertise, {gender_subj} was later recruited as director at {organization_3}. |
| | Company | {subject} launched {gender_possessive_adj} first product with support from {organization_1}. {Gender_subj} later collaborated on a major project with {organization_2}. Eventually, {subject} was acquired by {organization_3}. |
| Creative Work | Person | {subject} discovered a passion for creative work after encountering {creative_work_1}. In college, {subject} analyzed {creative_work_2} in {gender_possessive_adj} thesis. Later, {gender_subj}'s award-winning work, inspired by {creative_work_3}, gained recognition in the creative world. |
| | Company | {subject} built {gender_possessive_adj} culture on the influence of {creative_work_1}. Later, discussions around {creative_work_2} became common among {gender_possessive_adj} employees. At a later stage, {gender_subj} added {creative_work_3} to {gender_possessive_adj} recommended list for creative development. |

Table 11: All entities in `StoryPropagation`

| In-Domain / Out-of-Domain | Real-world Entity Type | Relation Template |
|---|---|---|
| In-Domain | Person | Martin Luther King Jr., Napoleon Bonaparte, William Wordsworth, William Shakespeare, Genghis Khan, Vincent van Gogh, Mother Teresa, Leonardo da Vinci, Eleanor Roosevelt, Theodore Roosevelt, Albert Einstein, Cleopatra VII, Frida Kahlo, Pablo Picasso, Rosa Parks, Elvis Presley, Joan of Arc, Franklin D. Roosevelt, Marie Antoinette, Henry VIII, Coco Chanel |
| | Language | Polish, Portuguese, English, Hindi, Swedish, German, Spanish, Turkish, Greek, Persian (Farsi), Hebrew, French, Arabic, Gujarati, Bengali, Dutch, Korean, Tamil, Telugu, Italian, Kazakh, Haitian Creole, Punjabi, Swahili |
| | Country | Iran, Malaysia, Colombia, Kenya, Armenia, Israel, Maldives, Vietnam, Saudi Arabia, Pakistan, Bangladesh, Turkey, Germany, Czech Republic, United States, Russia, Ukraine, Oman, Japan, South Korea, Belgium, Norway, New Zealand, Indonesia, Denmark, France, India, Spain, Iceland, Greece, Thailand |
| | Event | The Reign of Alexander the Great, The Fall of the Berlin Wall, The Spanish Conquest of the Aztecs, The Assassination of Julius Caesar, The Collapse of the Soviet Union, The Battle of Midway, The Surrender of Japan in WWII, Abolition of Slavery in the US, The Establishment of the Ming Dynasty, The Emancipation Proclamation, The Execution of King Louis XVI, The Partition of India and Pakistan, The Assassination of John F. Kennedy, Signing of the Magna Carta, American Civil War, Moon Landing, The Battle of Thermopylae, The Establishment of the People's Republic of China, Fall of Constantinople, The Founding of the United States of America, The Taiping Rebellion, The Vietnam War, The Battle of Waterloo, Civil Rights Movement |
| | Organization | Toyota, Human Rights Watch, Sony, Spotify, The Salvation Army, Amazon, Bill & Melinda Gates Foundation, Apple, The ACLU, Ford, World Food Programme, Amnesty International, Siemens, Johnson & Johnson, World Health Organization, Nestlé, Alibaba, Airbnb, Walmart
What primary service or product does {organization} provide? |
| | Species | pygmy hippo, panda, praying mantis, red-shouldered hawk, swan, humpback whale, crocodile, snow leopard, tiger, king cobra, great horned owl, great white shark, wolverine, bengal tiger, whale shark, bald eagle, wildebeest, harpy eagle |
| | Creative Work | The Brothers Karamazov, Oldboy, The Count of Monte Cristo, Jane Eyre, Citizen Kane, The Hobbit, Gangnam Style, A Tale of Two Cities, War and Peace, Goodfellas, The Dark Knight, Brave New World, Catch-22, Pulp Fiction, The Grapes of Wrath |
| Out-of-Domain | Person | Alexander the Great, Machiavelli, Charles Dickens |
| | Language | Afrikaans, Sinhala, Russian, Malay, Ukrainian |
| | Country | Portugal, Italy, Sweden, Netherlands, Poland, Azerbaijan, Hungary |
| | Event | The Boston Tea Party, The Montgomery Bus Boycott, Protestant Reformation, The Haitian Revolution, Napoleonic Wars, French Revolution, The 9/11 Attacks, English Civil War, The Battle of Hastings |
| | Organization | Walt Disney Company |
| | Species | albatross, raccoon, mantis shrimp, giant panda, giraffe, sloth, chameleon |
| | Creative Work | Pride and Prejudice, The Road, A Separation, Spirited Away, Pan's Labyrinth |

Table 12: All relations in `StoryPropagation`

| In-Domain / Out-of-Domain | Real-world Entity Type | Relation Template |
|---|---|---|
| In-Domain | Person | What occupation is {person} most well-known for? |
| | | Where was the birthplace of {person}? |
| | | What language was primarily spoken by {person}? |
| | | What year did {person} pass away? |
| | | What is the religion of {person}? |
| | | What year was {person} born? |
| | Language | What writing system is used by {language}? |
| | | What is the ISO 639-1 code for {language}? |
| | | What region is {language} native to? |
| | Country | What is the top-level internet domain for {country}? |
| | | What is the currency of {country}? |
| | | What is the ISO alpha-2 code for {country}? |
| | | Which ethnic group is the largest in {country}? |
| | | What is the capital of {country}? |
| | | What language in {country} has the most speakers? |
| | | What is the calling code for {country}? |
| | Event | In which country did {event} happen? |
| | | Who was the most important leader or figure involved in {event}? |
| | Organization | Where was {organization} established? |
| | | In what year was {organization} established? |
| | | Who established {organization}? |
| | | What is the primary field or industry of {organization}? |
| | | What primary service or product does {organization} provide? |
| | Species | What is the social structure of {species}? |
| | | What is the diet of {species}? |
| | | What type of organism is {species}? |
| | Creative Work | What is the original language of {creative_work}? |
| | | When was {creative_work} released or published? |
| | | Where was {creative_work} produced or created? |
| | | In which country was {creative_work} first released or published? |
| | | What is the genre or style of {creative_work}? |
| Out-of-Domain | Person | ∅ |
| | Language | What is the name of the alphabet or script of {language}? |
| | Country | Which religion has the most followers in {country}? |
| | Event | When did {event} take place? |
| | | What year did {event} end? |
| | Organization | Where is the headquarters of {organization} located? |
| | Species | Where is {species} primarily native to? |
| | Creative Work | Who is the creator of {creative_work}? |

Table 13: Hyperparameters used for Continue Pretraining baselines, CPT (Full) and CPT (Mid-Upper)

| Hyperparamter | Value |
|---|---|
| Learning rate | 1e-5 |
| Scheduler | linear |
| Epoch | 4 |
| Max seq. length | 1024 |
| Batch size | 1 |
| Weight decay | 0.1 |
| Max Gradient Norm | 1.0 |
| Optimizer | AdamW |

Table 14: Hyperparameters used for `PropMEND` and MEND.

(a) Hyperparameters for training `PropMEND` and MEND.

| Hyperparameter | Value |
|---|---|
| $c_{edit}$ | 0.1 |
| learning rate to learn test-time learning rate $\alpha_\ell$ | 0.0001 |
| Learning rate for hypernetwork weight $\phi$ | 1.0e-06 |
| Batch size (after gradient accumulation) | 10 |
| Validation step | 100 |
| Early stop patience (# steps) | 2000 |
| Maximum training step | 1000000 |
| Optimizer | Adam |

(b) Hyperparameters for hypernetwork (MLP) in `PropMEND` and MEND.

| Hyperparameter | Value |
|---|---|
| Activation | ReLU |
| # hidden | 1 |
| # hidden dim | 1920 |
| # parameter sharing | False |

(c) Target MLP layers used for various comparison system

| Base Model | Total # layers | Comparison system | Layer indices (min: 0) |
|---|---|---|---|
| `Llama-3.2-1B-base` | 16 | `PropMEND` | 4-15 |
| | | `PropMEND` (Mid-Upper) / MEND (Mid-Upper) | 10-12 |
| `Qwen2.5-1.5B-base` | 28 | `PropMEND` | 13-27 |
| `Llama-3.2-3B-base` | 28 | `PropMEND` | 15-27 |

propagation by testing whether the model is able to conduct multi-hop reasoning. In our preliminary study, we also considered a multi-hop question answering dataset for our study, but we found 100% verbatim rate from instances in MQuake-Remastered. A similar issue exists in MuSiQue [39] and other multi-hop question answering datasets [41]. Onoe et al. [32, 31] study the task of learning a new entity through description (e.g., "*Dracula*"), and ask inference questions about the entity (e.g., "Dracula makes you *fear*"). CodeUpdateArena [23] tests whether the model could learn a function update in the docstring difference and apply the updated function in program synthesis. ECLeKTic [10] focuses on cross-lingual knowledge transfer.

# G   Computational resources

We conducted experiments with `Llama-3.2-1B-base` primarily on a server with NVIDIA A40 48GB GPUs and an AMD EPYC 7413 24-Core Processor. For larger models, our experiments were conducted on a server with NVIDIA GH200 120GB and ARM Neoverse-V2.

Table 15: **Exact Match (EM) Results on** `RippleEdit`. We report the total number of test queries in brackets. Prepend is not a parametric method. The other metric (LLM-Acc) is reported in Table 1 in the main paper.

| EM (↑) | Efficacy | | Specificity | |
| --- | --- | --- | --- | --- |
| | Verbatim | Non-Verbatim | Verbatim | Non-Verbatim |
| | (1373) | (1586) | (165) | (2099) |
| `Llama-3.2-1B-base-QA` | 17.0 | 4.0 | 90.9 | 23.2 |
| + Prepend | 36.0 | 12.4 | 94.5 | 21.6 |
| + CPT (Full) | 87.8 | 3.4 | **99.4** | 17.3 |
| + CPT (Mid-Upper) | 48.7 | 4.0 | 93.3 | 24.1 |
| + MEMIT (`wikitext-103`) | 21.1 | 5.6 | 93.3 | 24.1 |
| + MEMIT (`RippleEdit`) | 26.6 | 5.9 | 98.2 | 19.3 |
| + MEND (with standard config) | 72.7 | 3.0 | 98.2 | 21.3 |
| + MEND (Mid-Upper) | 69.7 | 3.1 | 97.0 | 17.8 |
| + PropMEND (Mid-Upper) | 73.8 | 14.9 | 97.6 | 31.8 |
| + PropMEND | **78.7** | **17.3** | 95.2 | **35.1** |

Table 16: **Results on** `RippleEdit`. Performances are reported in the format of Exact Match (EM) / LLM-Accuracy. We notice the EM and LLM-Acc strongly disagree with each other on Forgetfulness (FN); after spotchecking, we found EM is high because one of the valid answers $a \in \mathcal{A}_i$ is a substring of the propagation question $q_i$. Prepend is not a parametric method.

| EM / LLM-Acc (↑) | Efficacy | | | | Specificity | |
| --- | --- | --- | --- | --- | --- | --- |
| | LG | CI | CII | SA | RS | FN |
| | (230) | (1679) | (273) | (777) | (1982) | (282) |
| `Llama-3.2-1B-base-QA` | 13.0/13.5 | 13.0/11.0 | 4.4/9.3 | 4.6/8.2 | 24.9/29.0 | 51.1/10.4 |
| + Prepend | 20.0/31.7 | 21.1/24.6 | 18.3/21.8 | 30.9/38.5 | 23.3/38.5 | 52.5/13.3 |
| + CPT (Full) | 16.1/11.4 | 12.7/10.4 | 93.8/89.3 | 97.0/93.0 | 19.9/17.8 | 47.5/3.3 |
| + CPT (Mid-Upper) | 13.9/15.8 | 13.3/12.0 | 32.6/32.2 | 50.1/51.7 | 26.4/28.0 | 48.6/10.9 |
| + MEMIT (`wikitext-103`) | 14.3/13.8 | 14.5/14.6 | 7.3/11.6 | 10.6/16.2 | 24.1/26.3 | 49.6/7.9 |
| + MEMIT (`RippleEdit`) | 14.3/13.3 | 14.8/14.8 | 7.7/13.9 | 20.2/24.9 | 21.6/23.5 | 48.9/7.3 |
| + MEND (with standard config) | 14.8/11.7 | 12.1/10.2 | 68.9/69.8 | 79.9/80.8 | 24.0/25.8 | 47.5/8.4 |
| + MEND (Mid-Upper) | 13.5/13.8 | 12.4/10.8 | 59.0/64.1 | 77.9/79.2 | 20.1/23.6 | 47.5/8.1 |
| + PropMEND (Mid-Upper) | 27.0/12.8 | 22.9/25.9 | 72.5/74.3 | 77.7/79.3 | 33.3/33.1 | 59.9/21.5 |
| + PropMEND | 30.9/25.0 | 25.3/27.7 | 83.5/85.7 | 81.3/82.1 | 35.7/35.6 | 65.6/27.3 |

Table 17: Results on `StoryPropagation` with `Qwen-2.5-1.5B-base-QA-FMT`. We use the model's LLM-Acc on alias questions for efficacy, and the model's performance on unalias questions for specificity. OOD (Entity) means using ID relation with OOD entity; OOD (Relation) means using ID entity with OOD relation. Prepend is not a parametric method.

| LLM-Acc (↑) | In-Domain | | OOD (Entity) | | OOD (Relation) | | OOD (Both) | |
| --- | --- | --- | --- | --- | --- | --- | --- | --- |
| | (2284) | | (1368) | | (421) | | (447) | |
| | Effi. | Spec. | Effi. | Spec. | Effi. | Spec. | Effi. | Spec. |
| `Qwen-2.5-1.5B-base-QA-FMT` | 8.0 | 91.2 | 6.8 | 89.9 | 10.5 | 87.3 | 9.1 | 91.1 |
| + Prepend | 66.9 | 88.3 | 64.9 | 87.8 | 60.3 | 84.1 | 55.5 | 83.3 |
| + CPT (Full) | 12.0 | 88.2 | 9.6 | 86.8 | 12.0 | 82.7 | 11.2 | 82.0 |
| + PropMEND | 64.3 | 93.4 | 34.1 | 80.2 | 34.5 | 83.4 | 16.7 | 82.8 |

Table 18: Results on `StoryPropagation` with `Llama-3.2-3B-base-QA-FMT`. We use the model's LLM-Acc on alias questions for efficacy, and the model's performance on unalias questions for specificity. OOD (Entity) means using ID relation with OOD entity; OOD (Relation) means using ID entity with OOD relation. Prepend is not a parametric method.

| LLM-Acc (↑) | In-Domain (2284) | | OOD(Entity) (1368) | | OOD(Relation) (421) | | OOD(Both) (447) | |
|---|---|---|---|---|---|---|---|---|
| | Effi. | Spec. | Effi. | Spec. | Effi. | Spec. | Effi. | Spec. |
| `Llama-3.2-3B-base-QA-FMT` | 8.1 | 91.8 | 6.9 | 93.0 | 8.1 | 92.4 | 6.5 | 93.8 |
| + Prepend | 69.8 | 91.8 | 68.4 | 92.9 | 64.1 | 92.0 | 56.6 | 94.3 |
| + CPT (Full) | 18.4 | 86.2 | 16.8 | 86.0 | 16.1 | 86.7 | 12.7 | 82.7 |
| + PropMEND | 69.9 | 94.6 | 42.4 | 89.8 | 34.0 | 93.2 | 19.2 | 89.6 |

Though the runtime varies depending on the datasets, the meta-training of hyper networks typically takes around 10 hours, or as little as 4 hours for some experiments.

