# OpenReview forum: "PropMEND: Hypernetworks for Knowledge Propagation in LLMs"
_NeurIPS.cc/2025/Conference — Submitted to NeurIPS 2025_

### Official Review · Reviewer_4bdp · 2025-06-24

**Clarity:** 2
**Significance:** 3
**Originality:** 2
**Rating:** 3
**Confidence:** 3

**Summary:**

Existing knowledge editing methods primarily enable verbatim knowledge injection, exhibiting limited efficacy in knowledge propagation. To address this limitation, the paper introduces PropMEND, a new knowledge propagation method. The paper further presents StoryPropagation, a dataset focusing on entities and relations that the model already understands well. Extensive experiments on StoryPropagation and RippleEdit datasets demonstrate that PropMEND achieves substantial performance improvements compared with baseline methods.

**Questions:**

1.Can the authors provide experimental results for one or more knowledge editing methods on the StoryPropagation dataset?
2.Will modifying the editing loss improve the other knowledge editing method 's effectiveness on the StoryPropagation dataset? Can the authors provide some experiment?
3.Will the test results of LLM acc be higher than those of EM?

**Ethical Concerns:**

["NO or VERY MINOR ethics concerns only"]

**Final Justification:**

After reading through the reviewers'comments and the corresponding rebuttals, I regret to say that the concerns on novelty still have not been well solved, which means the quality of this paper doesn't match the top-tier venues like NeurIPS.  I would keep my negative rating.

**Limitations:**

yes

**Quality:**

3

**Strengths And Weaknesses:**

Strengths:
1.New problem.This paper finds that existing methods excel primarily when target answers appear verbatim in edited facts, while demonstrating negligible improvement on non-verbatim queries.
2.New method.Conpared with mend, Propmend modifies the inner-loop loss function, outer-loop loss function, data input pipeline, and edit-layer decision, achieving significant performance gains on the StoryPropagation and RippleEdit benchmark datasets.
3.New dataset. StoryPropagation is a comprehensive dataset that includes 7 types of entities, allowing us to evaluate out-of-domain settings in knowledge propagation(Entity、relation、both)

Weaknesses：
1.Despite the paper's title emphasis on "knowledge propagation," both the proposed methodology and evaluation dataset exclusively address multi-hop question answering which only is a kind of knowledge propagation. This renders the broad title claim potentially misleading.
2.The baseline comparison remains limited, as the paper evaluates only MEND and MEMIT methods. Other important knowledge editing techniques (e.g., SERAC、IKE...) are notably absent from the experimental analysis.
3. Propmend primarily modifies only two loss functions (inner-loop and outer-loop) and the data input relative to MEND and is a simple extension to mend, absent of innovation.

---

> ### Author Rebuttal · Authors · 2025-07-31
>
> > Despite the paper's title emphasis on "knowledge propagation," both the proposed methodology and evaluation dataset exclusively address multi-hop question answering which only is a kind of knowledge propagation. This renders the broad title claim potentially misleading.
>
> We believe that propagating knowledge along knowledge base relations is an important propagation type, frequently studied in prior work [1]. Techniques developed for this task could generalize to other types of propagation, such as logical implications of a fact using world knowledge (“Barack Obama was president of the United States” -> “Barack Obama was born in the United States”), as knowledge base relations are quite diverse.
>
> [1] Evaluating the Ripple Effects of Knowledge Editing in Language Models; TACL 2024, Roi Cohen, Eden Biran, Ori Yoran, Amir Globerson, Mor Geva
>
> ---
>
> > The baseline comparison remains limited, as the paper evaluates only MEND and MEMIT methods. Other important knowledge editing techniques (e.g., SERAC、IKE...) are notably absent from the experimental analysis.
>
> Thanks for suggesting relevant work! Our work focus on methods that create *parametric* (delta weight) to the model weight. This is a more challenging editing scenario, but makes using the updated model a lot simpler. The suggested baselines do not have parameter updates. IKE uses a few-shot prompt to bias model to provide updated answers given the new fact and the question. SERAC maintains a memory and bias model’s behavior at test time, but does not modify the base model parameters. We will make the distinction clear in the paper.
>
> ---
> > Propmend primarily modifies only two loss functions (inner-loop and outer-loop) and the data input relative to MEND and is a simple extension to mend, absent of innovation.
>
> Our contribution is that we spot a mismatch between the problem formulation and the objective that all knowledge editing methods optimize against. We consider it a pretty significant change to be able to edit based on a general sequence of tokens rather than QA pairs. We don’t think it’s obvious that the modifications we’ve made to this algorithm would work or that meta-learning can succeed on this task, compared to MEND’s task where the edit and test conditions are more closely aligned.
>
> ---
>
> > Can the authors provide experimental results for one or more knowledge editing methods on the StoryPropagation dataset?
>
> In our Table 2 (main paper), we provide results for  existing knowledge editing methods — CPT, MEMIT and MEND.
>
> ---
> > Will modifying the editing loss improve the other knowledge editing method 's effectiveness on the StoryPropagation dataset?
>
> Possibly, but it’s not obvious to us how to make simple changes to other methods and achieve the same effect. Methods like MEMIT have a strict formulation of how its optimization problem is specialized for memorization-based knowledge editing, and adapting propagation questions to it is non-trivial.
>
> ---
> > Will the test results of LLM acc be higher than those of EM?
>
> Yes. LLM-Acc will always be higher than EM-Acc  since the LLM-Acc offers partial credit.

---

### Official Review · Reviewer_sqLb · 2025-07-03

**Clarity:** 4
**Significance:** 2
**Originality:** 2
**Rating:** 2
**Confidence:** 4

**Summary:**

How do you add a new fact into an LLM so that it is able to answer complex questions involving the new fact? There is an existing method called MEND (summary below) that backpropagates through a simple QA and its paraphrase in its inner and outer loops, respectively. This paper simply replaces the simple QA in the outer loop with complex QA of the kind that will be used during testing, and replaces conditional probability P ( A | Q) with generational probability P ( Q; A) in the inner loop.
It also introduces a new synthetic dataset (StoryPropagation) to address the limitation of the existing benchmark ( RippleEdit) after observing that many questions in RippleEdit involve tail entities that LLM may not already be aware of.

**Summary of MEND:** it trains a hypernetwork whose input is the gradient of the LLM w.r.t. the new fact, and the output is the modified gradient (or delta weights) that should be added to the LLM weights. The major technical innovation of MEND lies in observing that gradients of a weight matrix W in a layer (e.g y = Wx) is actually rank 1 ( $del L / del W = (del L / del y)*(x^T)$, and the hypernetwork can be designed to exploit that. Instead of taking $del L / del W$ as input, which would be $O(d^2)$, it takes, $concat(del L / del y ; x)$  as input, which is $(O(2d))$.

**Questions:**

1.  In Table 6 (a), you show a snapshot of world knowledge. Do you expect the model to already know this before you make the edit?

2. From the example in Table 6(a), it looks like it is an addition and not an edit, as sibling is a many-to-many relationship. How does it work in the case of knowledge “edit” - ie, old fact is no longer true, e.g. updating the current PM of a nation.

3. Line 206-207: “Many questions involve tail entities, where the base LM is not equipped with the information”: I think FMT training in StoryPropagation takes care of this issue as well.  Why can’t you do the same for RippleEdit? I.e. do SFT with QA that represents the world knowledge (e.g. Fig. 6(a))

4. Line 229: “we further align the model’s format on the generated question-answer pairs”: why do you call it alignment to format and not ingesting or reinforcing the knowledge required to answer the question?

5. Line 237: Why not use existing QA datasets like NQ or musique (for testing multihop reasoning) to test OOD generalisation?

6. Table 2: Why is the row for PropMend bold in OOD (Entity) Effi. and OOD (Both) Effi. columns?

7. Table 4: propagation --> paraphrases: Are you replacing multihop QA with their paraphrases, or just the paraphrase of the input fact to be edited? If it is the latter, then isn’t it an unfair comparison?

**Typos:**

line 253: “degradation in out-of-domain settingsWhen”

line 269: “Table 2 presents”: I think you want to cite table 4.

line 561: ““achieves an accuracy than 0.4.”

**Ethical Concerns:**

["NO or VERY MINOR ethics concerns only"]

**Final Justification:**

I have gone through the other reviews and the rebuttal. My concern regarding the novelty and applicability of the technique to edit only a limited number of facts still remains. I would prefer to keep my rating.

**Limitations:**

yes

**Paper Formatting Concerns:**

No formatting concerns

**Quality:**

2

**Strengths And Weaknesses:**

**Strengths**

1. The paper aims to solve a very pertinent problem -- how to use the newly acquired information to answer complex reasoning questions involving the newly gathered information.

2. The paper is well written and easy to understand.

3. It introduces a new dataset that can be used as a benchmark to evaluate other methods.

**Weaknesses**

1.  My major concern with the paper is in its little or incremental technical contribution -- all it is suggesting is to replace the training data in MEND with the kind of data that will be used during testing -- it is obvious and bound to give improvement. The other change is to replace the conditional probability P( Y | X) during gradient computation with generational probability P ( X; Y) - is this just an empirical observation, or is it necessitated by the lack of data in the form of QA pairs,  or is there any theoretical justification for it?

2.  My next concern is with experiments --

       a.  Why are the experiments conducted with the base model (Llama-3.2-1B-base) and not the corresponding instruct version (Llama-3.2-1B-instruct)? Will this not circumvent the need for training with TriviaQA?

       b.  **Stronger baselines** - Methods like MEND and PropMEND use the training data to learn to prepare the model to adapt to the new facts at test time.  There are works that propose creating synthetic data to prepare the model to adapt to new facts at test time via CPT (e.g. Self-Tuning (http://arxiv.org/abs/2406.06326). A simple way to adapt it to your use case would be to do CPT+SFT using your training data and then do CPT with the new fact at test time. Instead of using Llama-3.2-1B-base-QA in your “CPT” and “prepend” baselines, I think you should use a model trained using your training data via CPT+SFT (with or without TriviaQA dataset).


3. Finally, what happens when you have to ingest multiple stories in a single LLM? MEND shows that it could absorb up to 125 facts in a single LLM by simply adding the weight changes.
Relatedly, it seldom happens that we have to edit just a few facts or knowledge snippets in an LLM. Often, the new knowledge would be presented in the form of plain text, without access to any complex questions or even simple questions. Does PropMEND (or even MEND) require converting plain text into atomic facts and multihop questions involving those facts? Even then, the limitation of its ability to absorb only a handful of edits in a single LLM remains.

---

> ### Author Rebuttal · Authors · 2025-07-31
>
> **Due to space limitations, we focus on addressing the most important concerns. If the reviewer still has questions regarding other points in "Questions" section, we are more than happy to address them!**
>
> > My major concern with the paper is in its little or incremental technical contribution ... is there any theoretical justification for it?
>
> We agree that we make relatively small modifications to the existing MEND architecture, which is why we named our approach “PropMEND”! But our experiments show these  modifications bring meaningful differences in performance. Furthermore, it allows the framework to be used in more general settings, where facts cannot be easily reformulated into QA formats. We also introduce a scenario where the target task (the outer loop of MEND) differs substantially from the training task (the inner loop of MEND), making the meta-learning problem harder, and shows the method can work well in this scenario as well, which was not tested previously.
> Our contribution is a new problem framing, evaluation framework (which allows OOD evaluation and targeted knowledge propagation), as well as an effective solution that adapts an existing method.
>
> ---
>
> > My next concern is with experiments --  Why are the experiments conducted with the base model (Llama-3.2-1B-base) and not the corresponding instruct version (Llama-3.2-1B-instruct)? Will this not circumvent the need for training with TriviaQA?
>
> It is easier to inject knowledge to base models when the learning rate is still high. Prior work also uses a base model to study the learning of factual knowledge [1,2,3].  Anecdotally, it is difficult to inject knowledge into a model (via CPT or other means) after a long instruction training phase; this only works well if you do a longer training run with warmup and cool-down stages. We include evidence below. We include results when conducting a Continued Pretraining (CPT) experiment starting from Llama-3.2-1B-Instruct model (same experiment setup as Table 2)
>
> |                       |          | In-Domain |             | Out-of-Domain (Entity) |             | Out-of-Domain (Relation) |             | Out-of-Domain (Both) |             |
> |-----------------------|----------|-----------|-------------|------------------------|-------------|--------------------------|-------------|----------------------|-------------|
> |                       | Method   | Efficacy  | Specificity | Efficacy               | Specificity | Efficacy                 | Specificity | Efficacy             | Specificity |
> | Llama-3.2-1B-QA-FMT   | CPT      | 18.1      | 80.2        | 17.0                   | 79.9        | 15.6                     | 79.3        | 12.9                 | 71.1        |
> |                       | PropMEND | 76.7      | 95.5        | 35.2                   | 81.6        | 34.5                     | 84.0        | 18.3                 | 77.5        |
> | Llama-3.2-1B-Instruct | Base     | 0.5       | 80.6        | 0.3                    | 78.4        | 0.3                      | 83.9        | 0.5                  | 88.7        |
> |                       | CPT      | 1         | 76.4        | 0.7                    | 75.3        | 0.8                      | 78.8        | 0.5                  | 79.5        |
> |                       | PropMEND | 45.7      | 87.1        | 35.5                   | 84.1        | 26.3                     | 83.1        | 18.1                 | 84.9        |
>
> To share a bit of insight about why Llama-3.2-1B-Instruct model performs so badly, we share a type of common output from the model below:
> ```
> I couldn't find any information on an individual named Eric Ward being recruited as director at a specific organization
> ```
> We follow prior work [1,2,3] in this domain, which choose base models to edit, as they can provide a cleaner testbed for studying knowledge injection without conflating it with how a model is aligned.
>
> [1] How Do Large Language Models Acquire Factual Knowledge During Pretraining?; NeurIPS 2024, Hoyeon Chang, Jinho Park, Seonghyeon Ye, Sohee Yang, Youngkyung Seo, Du-Seong Chang, Minjoon Seo
>
> [2] Memorization without overfitting: Analyzing the training dynamics of large language models; NeurIPS 2022 Kushal Tirumala, Aram H. Markosyan, Luke Zettlemoyer, Armen Aghajanyan
>
> [3] Dissecting recall of factual associations in auto-regressive language models; EMNLP 2023, Mor Geva, Jasmijn Bastings, Katja Filippova, Amir Globerson
>
> ---
> > Stronger baselines ....
>
> Thanks for suggesting a relevant work!
>
> Self-tuning isn’t quite straightforward to work with in our present data condition. The authors of Self-tuning haven’t released scripts they use to create the memorization/comprehension/self-reflection examples, so there isn’t a plug-and-play experiment that’s doable with this method. We were not able to reimplement their method during the rebuttal period. We conducted an experiment in a similar vein to Self-tuning: we conduct SFT training on propagation questions in a meta-training set, and run CPT on each injected fact. As shown in the first row in the table below, the additional SFT only mildly improves efficacy performance but damages specificity performance. (**See "CPT (after SFT on propagation question)" in Table below**)
>
> We also note that the suggested Self-Tuning method only improves marginally over the CPT performance in their work. On StoryPropagation, CPT baseline provides a relatively small scale of improvement, from 8.3% to 18.1%. If we extrapolate a similar performance improvement over CPT, the performance of Self-Tuning will still be far below that of PropMEND.
>
> ---
> > Finally, what happens when you have to ingest multiple stories ....
>
> We experimented with doing 5 and 10 edits at the same time, and training our hypernetwork for these settings. The  performance degrades compared with the single-edit scenario, but we still outperform our baselines *even when those baselines use single-edits* (Table 2).  To accommodate more edits, we believe a larger hypernetwork or different learning rate would be helpful, which is doable but requires some additional engineering and efficiency improvements beyond the scope of the current work.
>
> |                                         | In-Domain |             | OOD (Entity) |             | OOD (Relation) |             | OOD (Both) |             |
> |-----------------------------------------|-----------|-------------|--------------|-------------|----------------|-------------|------------|-------------|
> |                                         | Efficacy  | Specificity | Efficacy     | Specificity | Efficacy       | Specificity | Efficacy   | Specificity |
> | Base                                    | 8.3       | 94.7        | 7.1          | 94.3        | 8.9            |  94.2       | 10.9       | 90.7        |
> | CPT (after SFT on propagation question) | 24.9      | 79.3        | 14.7         | 49.0        | 13.0           | 56.5        | 12.9       | 54.5        |
> | CPT                                     | 18.1      | 80.2        | 17.0         | 79.9        | 15.6           | 79.3        | 12.9       | 71.1        |
> | MEMIT (wikitext-103)                    | 12.8      | 94.4        | 14.4         | 94.4        | 12.0           | 93.9        | 13.8       | 90.0        |
> | PropMEND # edit = 1                     | 76.7      | 95.5        | 35.2         | 81.6        | 34.5           | 84.0        | 18.3       | 77.5        |
> | # edit = 5                              | 55.0      | 93.8        | 29.0         | 83.0        | 21.7           | 88.2        | 15.4       | 87.2        |
> | # edit = 10                             | 25.5      | 68.2        | 15.5         | 48.4        | 10.9           | 54.6        | 11.4       | 56.5        |
>
> ---
>
> > Line 206-207: “Many questions involve tail entities,... I.e. do SFT with QA that represents the world knowledge (e.g. Fig. 6(a))
>
> Great question. There is a distinction between a fact that the model has been exposed to at pre-training time [3], typically in various forms, and one that it hasn’t. Fine-tuning on a fact the model has seen before “resurfaces” that fact in a way that’s different from fine-tuning at instruction time. [1] shows that finetuning pretrained model on unfamiliar knowledge (based on the pretrained model) will encourage hallucination. Related observations are noted in [2]. RippleEdit often features much more obscure knowledge and this approach is expected to be less effective there. We will clarify this point in any future version.
>
> [1] Does Fine-Tuning LLMs on New Knowledge Encourage Hallucinations? EMNLP 2024, Zorik Gekhman, Gal Yona, Roee Aharoni, Matan Eyal, Amir Feder, Roi Reichart, Jonathan Herzig
>
> [2] Understanding Finetuning for Factual Knowledge Extraction; ICML 2024, Gaurav Ghosal, Tatsunori Hashimoto, Aditi Raghunathan
>
> [3] How Do Large Language Models Acquire Factual Knowledge During Pretraining?; NeurIPS 2024, Hoyeon Chang, Jinho Park, Seonghyeon Ye, Sohee Yang, Youngkyung Seo, Du-Seong Chang, Minjoon Seo
>
> ---
>
> > Line 237: Why not use existing QA datasets like NQ or musique (for testing multihop reasoning) to test OOD generalisation?
>
> We initially conducted experiments with MuSiQue. However, the dataset does not test the same kind of propagation as StoryPropagation, where part of the propagation relationship is hidden. Suppose we have an instance from MuSiQue with two paragraphs A and B containing relevant facts. Injecting both A and B leads to issues of memorization, since the answer tokens are present in either A or B; a baseline of repeating likely tokens from these paragraphs does very well. If we inject just one of these to avoid this, it’s unclear whether the model knows the content of the other paragraph. StoryPropagation explicitly avoids these reasoning shortcuts while testing the same kinds of effects.

---

> > ### Author Response · Authors · 2025-07-31
> > **Rebuttal (Continued)**
> >
> > > Does PropMEND (or even MEND) require converting plain text into atomic facts and multihop questions involving those facts?
> >
> > Line 107-111: PropMEND does not require converting plain text into atomic facts, whereas MEND does.
> >
> > ---
> > > In Table 6 (a), you show a snapshot of world knowledge. Do you expect the model to already know this before you make the edit?
> >
> > Yes. It is commonly observed that a pretrained LLM can work as a knowledge base [1]. Therefore, we specifically design our StoryPropagation dataset to contain knowledge that the model definitely knows — well-known entities and relations.
> >
> > [1] Language Models as Knowledge Bases? EMNLP 2019 Fabio Petroni, Tim Rocktäschel, Patrick Lewis, Anton Bakhtin, Yuxiang Wu, Alexander H. Miller, Sebastian Riedel
> >
> > ---
> > > From the example in Table 6(a), it looks like it is an addition and not an edit, as sibling is a many-to-many relationship. How does it work in the case of knowledge “edit” - ie, old fact is no longer true, e.g. updating the current PM of a nation.
> >
> > We consider the notion of “editing” with respect to what models already know. Both adding or editing knowledge is in scope (we do not deal with the case of removal in this work). If the old fact is no longer true, then the model is expected to only give the new fact.
> >
> > ---
> > > Line 229: “we further align the model’s format on the generated question-answer pairs”: why do you call it alignment to format and not ingesting or reinforcing the knowledge required to answer the question?
> >
> > Yes, it will reinforce the knowledge as well. Thanks for the catch, we will change the wording.
> >
> > ---
> > > Table 2: Why is the row for PropMend bold in OOD (Entity) Effi. and OOD (Both) Effi. columns?
> >
> > Prepend baseline does not make a parametric update to the model weight (Line 177), and does not count as a comparable system. We include it there as a reference.
> >
> > ---
> > > Table 4: propagation --> paraphrases: Are you replacing multihop QA with their paraphrases, or just the paraphrase of the input fact to be edited? If it is the latter, then isn’t it an unfair comparison?
> >
> > The “paraphrase” experiment uses the paraphrase of the input fact. We disagree with the reviewer that this is an unfair comparison, since paraphrases of the input fact are frequently used in knowledge editing work [1,2,3,4]. Propagation questions involve additional dimensions of information, but we view these as a data augmentation scheme fundamentally similar to paraphrasing. Our contribution is that we spot a mismatch between the problem formulation and the objective that all knowledge editing methods optimize against, a small change which brings big performance improvements.
> >
> > [1] AlphaEdit: Null-Space Constrained Knowledge Editing for Language Models; ICLR 2024. Junfeng Fang, Houcheng Jiang, Kun Wang, Yunshan Ma, Shi Jie, Xiang Wang, Xiangnan He, Tat-seng Chua
> >
> > [2] Aging with GRACE: Lifelong Model Editing with Discrete Key-Value Adaptors; Thomas Hartvigsen, Swami Sankaranarayanan, Hamid Palangi, Yoon Kim, Marzyeh Ghassemi, NeurIPS 2023
> >
> > [3] Do Localization Methods Actually Localize Memorized Data in LLMs? A Tale of Two Benchmarks; Ting-Yun Chang, Jesse Thomason, Robin Jia, NAACL 2024
> >
> > [4] InstructEdit: Instruction-based Knowledge Editing for Large Language Models; Ningyu Zhang, Bozhong Tian, Siyuan Cheng, Xiaozhuan Liang, Yi Hu, Kouying Xue, Yanjie Gou, Xi Chen, Huajun Chen, IJCAI 2024

---

> > > ### Comment · Reviewer_sqLb · 2025-08-04
> > >
> > > Dear authors
> > >
> > > Thank you for a detailed rebuttal. My concerns raised in Weakness 2 are sufficiently addressed.
> > >
> > > ____
> > >
> > > For editing multiple facts (edit 5 and edit 10), it looks like you are training new hypernetworks. How about using the same hypernetwork multiple times ( the way the MEND paper does by adding all the updates)?
> > >
> > > ____
> > >
> > > > Does PropMEND (or even MEND) require converting plain text into atomic facts and multihop questions involving those facts?
> > > >> Line 107-111: PropMEND does not require converting plain text into atomic facts, whereas MEND does.
> > >
> > > Would you not require converting text into complex QA for the outer loop in PropMEND?
> > >
> > > Regards,

---

> > > > ### Author Response · Authors · 2025-08-05
> > > >
> > > > > For editing multiple facts (edit 5 and edit 10), it looks like you are training new hypernetworks. How about using the same hypernetwork multiple times ( the way the MEND paper does by adding all the updates)?
> > > >
> > > > We experimented with multiple edits in the style of MEND and found that performance degraded more rapidly in the StoryPropagation setting than MEND found in their setting. We suspect that this is because the propagation objective requires more substantial changes to the network than fact memorization as in MEND, and updates don’t compose as easily unless the model is trained for this setting.
> > > >
> > > > In any case, with PropMEND, there is no reason that we can’t optimize for the actual test-time setting, whether that’s a single-edit or higher setting.
> > > >
> > > >
> > > > ---
> > > > > Would you not require converting text into complex QA for the outer loop in PropMEND?
> > > >
> > > > We require propagation questions in the outer loop, but this is only required for the training stage. At inference time, PropMEND allows the user to only provide naturally occurring text, whereas the previous formulation as in MEND would require the user to provide facts in QA format.

---

### Official Review · Reviewer_UKUc · 2025-07-03

**Clarity:** 2
**Significance:** 2
**Originality:** 2
**Rating:** 4
**Confidence:** 3

**Summary:**

The paper presents a hypernetwork-based method, aiming to improve knowledge propagation after editing large language models (LLMs). Unlike past approaches that focus mainly on inserting knowledge such that it is verbatim reproducible, this method explicitly meta-learns to modify model gradients so that newly injected facts propagate to multi-hop, non-verbatim queries. The method extends the MEND framework by altering its meta-objective — training the hypernetwork to transform gradients from fact injection so as to help the model answer compositional, relational, and aliasing questions. The method shows marked gains for non-verbatim, reasoning-intensive queries over a range of baselines.

**Questions:**

a) The performance drops in OOD (Both) settings are substantial (), especially relative to the in-domain result (64%). Could the authors clarify what factors contribute most to this degradation? Is it primarily limited data exposure, inductive bias of the hypernetwork, or something else? Would curriculum training or meta-augmentation be promising future directions?

b) As the method and training focus on single-edits, could the authors provide either preliminary empirical results or a more detailed theoretical discussion about how the method might extend to multiple, possibly conflicting edits in sequence? Are there clear bottlenecks that would require new hypernetwork designs?

**Ethical Concerns:**

["NO or VERY MINOR ethics concerns only"]

**Final Justification:**

The rebuttal responses to my questions, and I remain positive about this paper.

**Limitations:**

yes

**Paper Formatting Concerns:**

Not found

**Quality:**

2

**Strengths And Weaknesses:**

Strength
a) The paper tackles an important problem in knowledge editing: going beyond verbatim memory to true reasoning and propagation.
b) The extension of MEND to a meta-objective tailored to propagation is well-argued.

Weakness
a) In the out-of-domain (OOD) settings of the synthetic dataset (Table 2: OOD (Entity), OOD (Relation), OOD (Both)), the performance gap narrows noticeably; the accuracy drops to 17% for hardest cases, compared to 64% in-domain. This raises the concern that the method may not flexibly generalize propagation to new relations/entities, limiting real-world applicability, and undermining the tagline of “propagation” as a generally solved problem.
b) The focus remains on single-edit scenarios. However, real-world LLM deployment often requires continual, multi-point knowledge updating. The method’s suitability for such cases is not empirically or theoretically addressed (Limitations section).

---

> ### Author Rebuttal · Authors · 2025-07-31
>
> > This raises the concern that the method may not flexibly generalize propagation to new relations/entities, limiting real-world applicability, and undermining the tagline of “propagation” as a generally solved problem
>
> Our approach improves over past work  both in methodology and evaluation, yet do not frame it as solving propagation.  Substantial past work on knowledge editing [1,2,3,4] studies datasets which are primarily about verbatim memorization, where the notions of generalization we study don’t apply.
>
> [1] AlphaEdit: Null-Space Constrained Knowledge Editing for Language Models; ICLR 2024. Junfeng Fang, Houcheng Jiang, Kun Wang, Yunshan Ma, Shi Jie, Xiang Wang, Xiangnan He, Tat-seng Chua
>
> [2] Aging with GRACE: Lifelong Model Editing with Discrete Key-Value Adaptors; Thomas Hartvigsen, Swami Sankaranarayanan, Hamid Palangi, Yoon Kim, Marzyeh Ghassemi, NeurIPS 2023
>
> [3] Do Localization Methods Actually Localize Memorized Data in LLMs? A Tale of Two Benchmarks; Ting-Yun Chang, Jesse Thomason, Robin Jia, NAACL 2024
>
> [4] InstructEdit: Instruction-based Knowledge Editing for Large Language Models; Ningyu Zhang, Bozhong Tian, Siyuan Cheng, Xiaozhuan Liang, Yi Hu, Kouying Xue, Yanjie Gou, Xi Chen, Huajun Chen, IJCAI 2024
>
> - --
>
> > The performance drops in OOD (Both) settings are substantial (), especially relative to the in-domain result (64%). Could the authors clarify what factors contribute most to this degradation? Is it primarily limited data exposure, inductive bias of the hypernetwork, or something else? Would curriculum training or meta-augmentation be promising future directions?
>
> We agree understanding *why* degradation happens in OOD scenarios is important. We also thought that limited data or model size of the hypernetwork can be a bottleneck.  Thus, we experimented with scaling the training data and the parameter size for hypernetwork, see the table containing new results below. While this helped the performance in all evaluation settings, OOD performance was still much lower than in-domain performance.  We think curriculum training, new hypernetwork architecture, or changing the training data are all valid directions for future research, but do not have strong evidence supporting one solution over the other.
>
> Data: Synthetically created training data from StoryPropagation knowledge triplets
>
> Model: Llama-3.2-1B-base-QA-FMT
>
> |          | Hypernetwork size (# Param) | # train instances | In-Domain |             | Out-of-Domain (Entity) |             | Out-of-Domain (Relation) |             | Out-of-Domain (Both) |             |
> |----------|-----------------------------|-------------------|-----------|-------------|------------------------|-------------|--------------------------|-------------|----------------------|-------------|
> |          |                             |                   | Efficacy  | Specificity | Efficacy               | Specificity | Efficacy                 | Specificity | Efficacy             | Specificity |
> | PropMEND | 159M                        | 4K                | 76.7      | 95.5        | 35.2                   | 81.6        | 34.5                     | 84.0        | 18.3                 | 77.5        |
> |          | 2.8B                        | 30K               | 97.8      | 97.1        | 42.5                   | 87.2        | 41.8                     | 89.5        | 20.9                 | 87.8        |
>
> ---
>
> > The focus remains on single-edit scenarios. However, real-world LLM deployment often requires continual, multi-point knowledge updating. The method’s suitability for such cases is not empirically or theoretically addressed (Limitations section).
>
> > As the method and training focus on single-edits, could the authors provide either preliminary empirical results or a more detailed theoretical discussion about how the method might extend to multiple, possibly conflicting edits in sequence? Are there clear bottlenecks that would require new hypernetwork designs?
>
> We experimented with doing 5 and 10 edits at the same time, and training our hypernetwork for these settings. The  performance degrades compared with the single-edit scenario, but we still outperform our baselines *even when those baselines use single-edits* (Table 2).  To accommodate more edits, we believe a larger hypernetwork or different learning rate would be helpful, which is doable but requires some additional engineering and efficiency improvements beyond the scope of the current work.  We think continual edits are an exciting but orthogonal line of work from ours.
>
>
> |                      | In-Domain |             | OOD (Entity) |             | OOD (Relation) |             | OOD (Both) |             |
> |----------------------|-----------|-------------|--------------|-------------|----------------|-------------|------------|-------------|
> |                      | Efficacy  | Specificity | Efficacy     | Specificity | Efficacy       | Specificity | Efficacy   | Specificity |
> | Base                 | 8.3       | 94.7        | 7.1          | 94.3        | 8.9            |  94.2       | 10.9       | 90.7        |
> | CPT (Full)           | 18.1      | 80.2        | 17.0         | 79.9        | 15.6           | 79.3        | 12.9       | 71.1        |
> | MEMIT (wikitext-103) | 12.8      | 94.4        | 14.4         | 94.4        | 12.0           | 93.9        | 13.8       | 90.0        |
> | PropMEND # edit = 1  | 76.7      | 95.5        | 35.2         | 81.6        | 34.5           | 84.0        | 18.3       | 77.5        |
> | # edit = 5           | 55.0      | 93.8        | 29.0         | 83.0        | 21.7           | 88.2        | 15.4       | 87.2        |
> | # edit = 10          | 25.5      | 68.2        | 15.5         | 48.4        | 10.9           | 54.6        | 11.4       | 56.5        |
>
> We do notice that scaling to a larger number of edits will cause the runtime to increase. We also think doubling parameters of the hypernetwork is not the most elegant way to alleviate performance degradation. We believe future work focus could study the mechanism of knowledge propagation and leverage the insights to design a more efficient method.
>
> In StoryPropagation, since each fact/story is surrounding a fake entity (e.g., Adam Jacobson) and efficacy question is specific to each fake entity, so we don’t expect knowledge conflict comes into play. However, we do expect multiple edits will introduce conflict in propagation. We believe our work lays the foundation for future exploration.

---

> > ### Comment · Reviewer_UKUc · 2025-08-05
> >
> > The rebuttal responses to my questions, and I remain positive about this paper.

---

### Official Review · Reviewer_Cw9L · 2025-07-23

**Clarity:** 3
**Significance:** 2
**Originality:** 2
**Rating:** 2
**Confidence:** 4

**Summary:**

This paper addresses the problem of knowledge propagation in large language models (LLMs) after knowledge editing. The proposed method PropMEND is built on top of a previous work called MEND and uses a hyper-network based approach to improve knowledge propagation. The key contribution is training the hypernetwork to modify gradients of a language modeling loss such that injected knowledge can be used to answer multi-hop questions that require reasoning with the new information. The method is evaluated on RippleEdit and a new synthetic dataset called StoryPropagation, showing significant improvements over existing approaches, particularly for non-verbatim propagation questions.

This paper makes a meaningful contribution to knowledge editing by addressing the propagation problem with a hyper-network based approach. The idea of aligning the training objective with the desired propagation behavior is interesting. However, the work is limited by its scope (small models, single edits) and the synthetic nature of the evaluation. While the results are promising, more work is needed to demonstrate the method's applicability to real-world knowledge editing scenarios and larger-scale models.

**Questions:**

1. What are some simple and obvious ways to make the hypernetwork more parameter-efficient? Can some form of knowledge distillation be used? How does the size of the hyper-network effect the performance?
2. How does the method's performance degrade as the number of sequential edits increases? Are there interactions between multiple edits that affect propagation?
3. Can you provide more analysis on why editing lower layers (mid-upper rather than top layers) is more effective for knowledge propagation?
4. While the synthetic datasets are useful, the paper would benefit from evaluation on more realistic knowledge editing scenarios, perhaps using naturally occurring facts and propagation questions from knowledge bases -- picking a open-source model released few days ago and editing it with the facts post-training date might be an interesting experiment to try.

**Ethical Concerns:**

["NO or VERY MINOR ethics concerns only"]

**Limitations:**

Yes

**Quality:**

2

**Strengths And Weaknesses:**

Strengths

1. The paper tackles an important limitation of current knowledge editing methods - their inability to propagate injected knowledge beyond verbatim reproduction. This is a genuine gap in the field where existing methods can inject facts but fail at enabling reasoning with those facts.
2. In terms of evaluation, the distinction between verbatim and non-verbatim questions is particularly valuable, revealing that existing methods primarily succeed only on verbatim cases.

Weaknesses

1. While effective, the core contribution is essentially a modification of the loss function in MEND's training procedure. Compared to more recent developments in knowledge editing that address diverse evaluation criteria and architectural innovations, the technical advancement is somewhat incremental.
2. The evaluation is restricted to relatively small models (1B-3B parameters) and single-edit scenarios. The authors acknowledge this limitation but don't provide sufficient analysis of how the approach might scale to larger models or multi-edit scenarios that are more representative of real-world applications.
3. As acknowledged in the limitation, the hypernetwork is as large as the edited language model, inheriting MEND's parameter efficiency limitations. While efficiency comparisons are provided, the method's practicality for large-scale deployment remains questionable, especially given the need to target multiple layers.
4.  The paper lacks detailed analysis of when and why the method fails, particularly in challenging scenarios. Understanding failure patterns across different types of knowledge and reasoning requirements would strengthen the contribution.

---

> ### Author Rebuttal · Authors · 2025-07-31
>
> > Compared to more recent developments in knowledge editing that address diverse evaluation criteria and architectural innovations, the technical advancement is somewhat incremental.
>
> Our contribution is that we spot a mismatch between the problem formulation and the objective that existing knowledge editing methods optimize against.  We don’t think it’s obvious that the modifications we’ve made to the MEND algorithm would work or that meta-learning can succeed on this task, compared to MEND’s task where the edit and test conditions are more closely aligned. For our study, we follow the conventional evaluation criteria from studies on knowledge propagation [1,2,3].
>
> [1] Propagating Knowledge Updates to LMs Through Distillation; NeurIPS 2023, Shankar Padmanabhan, Yasumasa Onoe, Michael J.Q. Zhang, Greg Durrett, Eunsol Choi
>
> [2] Deductive Closure Training of Language Models for Coherence, Accuracy, and Updatability; ACL-Findings 2024, Afra Feyza Akyürek, Ekin Akyürek, Leshem Choshen, Derry Wijaya, Jacob Andreas
>
> [3] CaKE: Circuit-aware Editing Enables Generalizable Knowledge Learners. Arxiv 2025. Yunzhi Yao, Jizhan Fang, Jia-Chen Gu, Ningyu Zhang, Shumin Deng, Huajun Chen, Nanyun Peng
>
> ---
>
> > The evaluation is restricted to relatively small models (1B-3B parameters) and single-edit scenarios
>
> Knowledge editing evaluation is more costly than many other LLM research topics, as each knowledge edit instantiates a new model. This experimental setting (i.e., using 1-3B parameter models) is used in substantial amounts of work over the past year [1,2,3]. We aren’t aware of any evidence that editing methods effective at the 1B-3B scale wouldn’t work at a larger scale as well. Additional work on GPU memory management is needed for this scaling (which holds for any hypernetwork-based method); this is technically possible and is an ingredient of our ongoing work but is orthogonal to the key technical claims of this work.
>
> [1] AlphaEdit: Null-Space Constrained Knowledge Editing for Language Models; ICLR 2024. Junfeng Fang, Houcheng Jiang, Kun Wang, Yunshan Ma, Shi Jie, Xiang Wang, Xiangnan He, Tat-seng Chua
>
> [2] Aging with GRACE: Lifelong Model Editing with Discrete Key-Value Adaptors; Thomas Hartvigsen, Swami Sankaranarayanan, Hamid Palangi, Yoon Kim, Marzyeh Ghassemi, NeurIPS 2023
>
> [3] Do Localization Methods Actually Localize Memorized Data in LLMs? A Tale of Two Benchmarks; Ting-Yun Chang, Jesse Thomason, Robin Jia, NAACL 2024
>
> [4] InstructEdit: Instruction-based Knowledge Editing for Large Language Models; Ningyu Zhang, Bozhong Tian, Siyuan Cheng, Xiaozhuan Liang, Yi Hu, Kouying Xue, Yanjie Gou, Xi Chen, Huajun Chen, IJCAI 2024
>
> ---
>
> > The paper lacks detailed analysis of when and why the method fails, particularly in challenging scenarios. Understanding failure patterns across different types of knowledge and reasoning requirements would strengthen the contribution.
>
> We agree providing analysis into failure patterns would be important!
>
> Our dataset is designed with a control of well-known entities and relations, and allows us to study Out-of-Domain generalization by composing propagation questions with OOD entities or OOD relations (or both). As we noted in the paper (Line 257-259), we observe that OOD (Entity) performance tends to be higher than OOD (Relation). This result helps characterize the effectiveness of the hypernetwork at different challenging cases where it may have to edit different parts of the model than it’s seen during meta-training time. Designing evaluation along other axes (e.g., categories of propagation relation, like the reviewer suggested) can be great future work. We will add discussion on different analyses.
>
> ---
> > What are some simple and obvious ways to make the hypernetwork more parameter-efficient?
>
> The hypernetwork (for a weight matrix) is a 2-layer MLP which consists of a learned down-projection and a learned up-projection. It’s possible to reduce the size of this MLP further, or use low-rank parameterizations of the matrices, but we believe this would introduce a size-accuracy tradeoff.
>
> ---
> > How does the method's performance degrade as the number of sequential edits increases? Are there interactions between multiple edits that affect propagation?
>
>
> Thanks for the suggestion! Below we present new results of doing 5 and 10 edits at the same time, and training our hypernetwork for these settings. We list two baselines (CPT (Full) and MEMIT, the strongest baseline method from our work); the table is comparable to the results of Table 2 in the paper. The performance degrades compared with the single-edit scenario, but we still outperform our baselines *even when those baselines use single-edits* (Table 2).  To accommodate more edits, a larger hypernetwork or different learning rate can be helpful, which is doable but requires engineering and efficiency improvements beyond the scope of the current work.
>
> |                      | In-Domain |             | OOD (Entity) |             | OOD (Relation) |             | OOD (Both) |             |
> |----------------------|-----------|-------------|--------------|-------------|----------------|-------------|------------|-------------|
> |                      | Efficacy  | Specificity | Efficacy     | Specificity | Efficacy       | Specificity | Efficacy   | Specificity |
> | Base                 | 8.3       | 94.7        | 7.1          | 94.3        | 8.9            |  94.2       | 10.9       | 90.7        |
> | CPT (Full)           | 18.1      | 80.2        | 17.0         | 79.9        | 15.6           | 79.3        | 12.9       | 71.1        |
> | MEMIT (wikitext-103) | 12.8      | 94.4        | 14.4         | 94.4        | 12.0           | 93.9        | 13.8       | 90.0        |
> | PropMEND # edit = 1  | 76.7      | 95.5        | 35.2         | 81.6        | 34.5           | 84.0        | 18.3       | 77.5        |
> | # edit = 5           | 55.0      | 93.8        | 29.0         | 83.0        | 21.7           | 88.2        | 15.4       | 87.2        |
> | # edit = 10          | 25.5      | 68.2        | 15.5         | 48.4        | 10.9           | 54.6        | 11.4       | 56.5        |
>
>
> We don’t expect explicit conflicts of injected knowledge in the setting we consider, but we do expect multiple edits might require a higher-capacity hypernetwork to perform all of the necessary propagation. We believe our work lays the foundation for future exploration. We will add discussion and report results on multiedit scenarios in the revised paper.
>
> ---
>
> > Can you provide more analysis on why editing lower layers (mid-upper rather than top layers) is more effective for knowledge propagation?
>
> According to papers in the interpretability literature [1,2], mid-upper layers are where semantic processing happens in LLMs. The very top layers are more responsible for the mechanics of next token prediction. [3] use mechanistic interpretability techniques to show middle and upper layers are important for multi-hop queries. These studies motivated us to explore editing different layers than the original MEND paper.
>
> [1] Studying Large Language Model Generalization with Influence Functions; arxiv, 2023, Anthropic team
>
> [2] Linguistic Knowledge and Transferability of Contextual Representations; NAACL 2019, Nelson F. Liu, Matt Gardner, Yonatan Belinkov, Matthew E. Peters, Noah A. Smith
>
> [3] Hopping Too Late: Exploring the Limitations of Large Language Models on Multi-Hop Queries. EMNLP 2024 Eden Biran, Daniela Gottesman, Sohee Yang, Mor Geva, Amir Globerson
>
> ---
>
> > While the synthetic datasets are useful, the paper would benefit from evaluation on more realistic knowledge editing scenarios, perhaps using naturally occurring facts and propagation questions from knowledge bases -- picking a open-source model released few days ago and editing it with the facts post-training date might be an interesting experiment to try.
>
> We agree! However, it is unfortunately difficult to devise a controlled experiment of this form. Facts added to knowledge bases may be present in text prior to their inclusion in the KBs; so without exhaustively searching the LLM pre-training data, it’s difficult to tell whether the model has actually been pre-trained on this fact. We opted for a more controlled setting (especially with respect to out-of-domain-ness of entities and relations) and reproducibility for future work.

---

> ### Comment · Area_Chair_2yTa · 2025-08-08
> **from your AC, reminder to respond to author's rebuttal**
>
> Hi reviewer Cw9L,
>
> Can you respond to author's rebuttal? It is mandatory, and reviewer-author discussion is closing out in 24 hours. Thanks.
>
> AC

---

### Comment · Area_Chair_2yTa · 2025-08-03
**Reminder for author-reviewer discussion**

Hi reviewers,

Thanks for reviewing the paper. Could you take a look at authors' response and reply? Thank you.

Yours,

AC

---

### Decision · Program_Chairs · 2025-09-17

**Decision:**

Reject

**Comment:**

The work presents PropMEND that built on top of MEND loss function. Its empirical gain is encouraging and the direction is interesting. However, the evaluation currently focuses on limited domains (such as SLM and single edits). The current version of the work would benefit from stronger technical contributions and in-depth analysis.